# Unveiling phonons in a molecular qubit with four-dimensional inelastic neutron scattering and density functional theory

E. Garlatti [1,2], L. Tesi[3,5], A. Lunghi [4], M. Atzori [3,6], D. J. Voneshen [1], P. Santini[2], S. Sanvito[4], T. Guidi [1✉], R. Sessoli [3✉] & S. Carretta [2✉]

Phonons are the main source of relaxation in molecular nanomagnets, and different mechanisms have been proposed in order to explain the wealth of experimental findings. However, very limited experimental investigations on phonons in these systems have been performed so far, yielding no information about their dispersions. Here we exploit state-of-the-art single-crystal inelastic neutron scattering to directly measure for the first time phonon dispersions in a prototypical molecular qubit. Both acoustic and optical branches are detected in crystals of $[VO(acac)_2]$ along different directions in the reciprocal space. Using energies and polarisation vectors calculated with state-of-the-art Density Functional Theory, we reproduce important qualitative features of $[VO(acac)_2]$ phonon modes, such as the presence of low-lying optical branches. Moreover, we evidence phonon anti-crossings involving acoustic and optical branches, yielding significant transfers of the spin-phonon coupling strength between the different modes.

[1] ISIS Facility, Rutherford Appleton Laboratory, Didcot OX11 0QX, UK. [2] Dipartimento di Science Matematiche, Fisiche e Informatiche, Università di Parma and UdR Parma, INSTM, Parco Area delle Scienze 7/A, 43124 Parma, Italy. [3] Dipartimento di Chimica U. Schiff, Università degli Studi di Firenze and UdR Firenze, INSTM, Via della Lastruccia 3, I50019 Sesto Fiorentino, Firenze, Italy. [4] School of Physics, CRANN and AMBER Trinity College, Dublin 2, Ireland. [5]Present address: Institute of Physical Chemistry, University of Stuttgart, Pfaffenwaldring 55, 70569 Stuttgart, Germany. [6]Present address: Laboratoire National des Champs Magnétiques Intenses (LNCMI) – CNRS, 25 rue des Martyrs, 38042 Grenoble, France. ✉email: tatiana.guidi@stfc.ac.uk; roberta.sessoli@unifi.it; stefano.carretta@unipr.it

The two main pillars of the current research in molecular magnetism are the possibilities of exploiting molecules as classical bits for high-density magnetic memories[1–5] or as quantum bits (qubits) for quantum information processing[6–12]. The use of molecular nanomagnets (MNMs) as classical bits relies on their slow relaxation of the magnetisation, which is undermined by spin-phonon interactions[13]. Molecular vibrations may also play an important role in determining the magnitude and temperature dependence of coherence times in molecular qubits[14], which need to be very long to implement quantum algorithms.

Despite their importance, very limited experimental investigations on phonons in MNMs have been performed so far. Indeed, only integrated information on their low-energy spectrum or Γ-point energies have been determined in some compounds through specific heat measurements or Raman and THz spectroscopy techniques[13,15–17]. As also pointed out in a recent critical perspective on the topic[18], the key to construct a reliable model of phonon-induced relaxation dynamics in MNMs is to have access to phonon dispersions. In fact, several mechanisms take part into relaxation dynamics of MNMs (direct, Orbach, Raman and quantum tunnelling processes[5,19–23]), and the interplay between them depends on the phonon spectrum. Since experimental results on phonon dispersions were not available so far, many relaxation theories on MNMs have relied on the simple Debye model[24–26]. Even if they have been successful in understanding nuclear magnetic resonance (NMR) or dynamical susceptibility signals in some systems[26–29], the fundamental limitations of the Debye model prevent a quantitative description of relaxation dynamics in the new generations of MNMs[5,14,16,20,30]. Moreover, phonon eigenvectors are necessary for the evaluation of spin-phonon coupling coefficients, which requires a full description of molecular vibrations, including atomic displacements due to accessible phonon modes. In particular, the possible presence of low-lying optical phonons producing sizeable intra-molecular distortions can be crucial for magnetic relaxation[31]. Furthermore, anti-crossings (ACs) between these low-lying optical branches and acoustic phonons, which can be experimentally detected only by measuring phonon dispersions, are also important for magnetic relaxation. While single molecules could be the long-term goal for applications, the majority of experimental studies on the relaxation dynamics of MNMs are performed in crystals or polycrystalline samples. In addition, the use of magnetically diluted single crystals[32] can bring advantages in quantum simulation protocols, where the ensemble measurements immediately yield expectation values of the observables. Hence, the combination of an experimental technique directly addressing phonon dispersions and eigenvectors of a MNM crystal with state-of-the-art ab initio calculations is the starting point to investigate the physics behind relaxation mechanisms and benchmark theoretical models.

Here we exploit the four-dimensional inelastic neutron scattering approach (4D-INS)[21,33] and the LET spectrometer[34] to directly measure for the first time phonon dispersions in a molecular qubit prototype. Indeed, INS is a very powerful technique to study phonons, because it enables a direct access to phonon eigenvalues and eigenvectors. Moreover, the recent advent of spectrometers combining the time-of-flight technique with position-sensitive detectors makes measuring the four-dimensional scattering function $S(\mathbf{Q}, \omega)$ in large portions of reciprocal space possible[8,21,35,36]. This experimental method is largely applied to study phonon dispersions in inorganic systems with extended structures[37–41], but it has never been applied to complex molecular crystals. This work focuses on VO-acetylacetonate ($[VO(acac)_2]$), a prototypical complex embedding the vanadyl (VO) unit, archetype of a new generation of

molecular qubits with long coherence times up to high temperatures[14,16,30]. The results of this challenging 4D-INS experiment are compared to density functional theory (DFT) calculations. Simulations reproduce important qualitative features of the 4D-INS data, such as the presence of low-lying optical modes and ACs between acoustic and optical phonons, providing insights on their effect on spin-phonon couplings.

## Results

**The [VO(acac)₂] molecular qubit.** VO-based molecules contain a penta-coordinated $V^{IV}$ metal centre lying above the basal plane of a distorted square pyramidal coordination geometry. The double bond of the vanadyl $VO^{2+}$ ion is much shorter than single V–O bonds, leading to a strong axial distortion of the ligand field acting on the metal centre. (It should be highlighted that, despite the $VO^{2+}$ unit is commonly described by a double bond, a triple bond has also been proposed[42,43] and supported by theoretical calculations[44]. The resulting splitting of the $V^{IV}$ $d$-orbitals yields a $d_{xy}$ orbital lying the lowest in energy, singly occupied and well separated from the other orbitals. This also quenches the orbital contribution to the ground state, making these $S = 1/2$ systems ideal candidates as molecular spin qubits. The large splitting between the ground state doublet and the first excited levels also ensures that no magnetic excitations are present in the energy range up to more than 1500 meV[44]. Recent studies on this family of compounds suggest that the rigidity of the VO moiety is also at the basis of their long quantum coherence times up to room temperature[14,30,45].

$[VO(acac)_2]$ is the simplest molecule among all vanadyl complexes investigated so far (see Fig. 1), thus it is an ideal model system to investigate phonons in MNMs from both computational and experimental point of views. $[VO(acac)_2]$ crystallises in the triclinic centrosymmetric space group, with the unit cell containing two molecules, symmetric with respect to the inversion centre. The absence of magnetic excitations and the characteristics of its low-energy phonon spectrum, as predicted by DFT calculations, make it possible to measure both acoustic and optical modes with neutron incident energies which ensure a sufficiently high resolution. Moreover, it is also possible to grow sufficiently large single crystals of $[VO(acac)_2]$ of several mm in size, required to perform INS experiments, with a high percentage of deuteration (ca. 98%).

**Unveiling phonons with 4D-INS.** We exploited the cold-neutron LET spectrometer at ISIS[34] to measure the four-dimensional scattering function $S(\mathbf{Q}, \omega)$ of $[VO(acac)_2]$ (more details about $S(\mathbf{Q}, \omega)$ in the Methods sections), since the remarkable features of

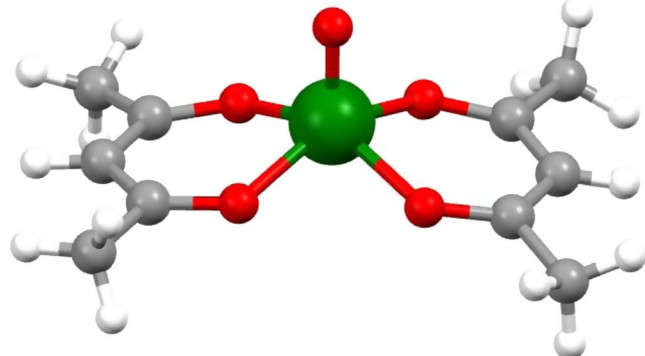

**Fig. 1 The [VO(acac)₂] molecular qubit.** Molecular structure of VO-acetylacetonate (green, V; red, O; grey, C; white, H). The double bond typical of the $VO^{2+}$ moiety is the apical one.

this instrument fit the requirements of this challenging experiment. Indeed, high-energy resolutions are required to resolve the phonon spectrum and, at the same time, it is important to explore wide portions of the reciprocal space. Moreover, we also need to measure at relatively small $Q$ even at high energies, because in this condition it is easier to focus on the coherent scattering signal. LET offers high-energy resolutions combined with position-sensitive detectors covering a wide solid angle, thus a wide $Q$ range. In addition, its high neutron flux and low background enable one to obtain high-quality data also from weakly scattering samples. Large crystals and a high percentage of deuteration are indeed required for these INS experiments, but these two requirements are not easily fulfilled in MNMs. The investigated sample was composed of 40 $[VO(acac)_2]$ co-aligned deuterated single crystals, in order to obtain a final mass of about 1 g (more details in the Methods sections). Measurements were performed at the temperature $T = 5$ K to minimise the phonon line broadening, with two different incident neutron energies $E_i = 7.3$ and 13 meV (see Methods). The $[VO(acac)_2]$ 4D scattering function $S(Q, \omega)$ was then reconstructed using the software HORACE[46], which allowed us to slice the 4D datasets into 1D curves to visualise excitations intensities as a function of energy around desired $Q$ values and into 2D slices along specified trajectories in the $(Q, E)$-space to visualise phonon dispersions along different directions.

An important feature of the INS technique is the possibility to experimentally distinguish between phonon modes with longitudinal or transverse polarisation, characterised respectively by lattice vibrations parallel or perpendicular to the direction of propagation. This is done by properly selecting the relative direction between the neutron scattering vector $Q$ and the phonon polarisation vector $\sigma_j^d(q)$, thanks to the dot-product in Eq. (4) (see Methods). For instance, longitudinal modes are extracted by inspecting data obtained with $Q$ vectors parallel to the desired $\sigma_j^d(q)$: e.g., longitudinal modes along Γ–Z are obtained by selecting $Q$ vectors of the type $(0, 0, Q_z)$. Conversely, $Q$-trajectories with sizeable components perpendicular to $\sigma_j^d(q)$ have been chosen to probe transverse phonon modes. Given the very low symmetry of $[VO(acac)_2]$, we do not expect phonon branches to be purely transverse or longitudinal (e.g., $\sigma_j^d(q)$ may not contain displacements only perpendicular or parallel to $Q$)[47]. Anyway, as far as acoustic branches are concerned, it is often convenient to call them "longitudinal" and "transverse", although they are not pure.

Phononic excitations intensities of $[VO(acac)_2]$ are reported as a function of energy in Fig. 2 for some representative $Q$ values along Γ–X, Γ–Y and Γ–Z symmetry directions, for both $E_i = 7.3$ meV (Fig. 2a–c) and 13 meV (Fig. 2d–g) incident energies. The former show one or two excitation peaks, due to longitudinal and transverse acoustic modes, while $E_i = 13$ meV data show phononic excitations up to 11 meV from both acoustic and optical modes.

1D cuts give a detailed insight onto excitations, whereas 2D slices of the 4D datasets provide an immediate visualisation of phonon dispersions, also demonstrating the capabilities of the 4D-INS technique. For instance, the data obtained with $E_i = 7.3$ meV in Fig. 3 provide a picture of $[VO(acac)_2]$ acoustic phonons dispersions up to 6 meV. Representative examples of the measured dispersion curves along Γ–Z, Γ–X and Γ–Y symmetry directions are reported in Fig. 3. Along Γ–Z, the longitudinal acoustic mode is the only intense one for data along the $[0, 0, \xi]$ longitudinal direction (Fig. 3a). Indeed, it corresponds to the steepest among the three modes along Γ–Z (see below). High-intensity modes in Fig. 3c and d are instead due to the two

transverse acoustic branches along Γ–X and Γ–Y, respectively, which reach the Brillouin zone boundary at only about 4 meV. In addition, low-lying optical modes are visible around 5 meV along the transverse Γ–Z direction reported in Fig. 3b. In order to visualise $[VO(acac)_2]$ low-energy optical modes, which could play an important role in magnetic relaxation, we also report phonon dispersions obtained with $E_i = 13$ meV in Fig. 4. Figure 4a confirms the longitudinal acoustic mode along Γ–Z shown in Fig. 3a and, in addition, is now possible to observe optical modes around 8 and 10 meV. Furthermore, optical modes up to 11 meV are clearly visible also in transverse configuration ($Qq$) along Γ–Z, Γ–X and Γ-Y directions (Fig. 4b–d, respectively).

Phonon ACs involve branches belonging to the same symmetry representation, which therefore cannot cross each other. In particular, INS data show that low-lying optical branches display ACs with acoustic ones in $[VO(acac)_2]$ (see Fig. 5). A strong mixing between the modes is expected close to ACs, which should significantly affect atomic displacements and thus crystal-field modulations involved in magnetic relaxation (see Discussion). Moreover, ACs cause a reduction of phonon lifetimes[48–50], again possibly affecting the relaxation dynamics[23]. We have therefore performed an analysis of acoustic-optical phonon ACs, considering transverse Γ–Z and Γ–X directions data. Results of this analysis are reported in Fig. 5. The position and intensity of each phononic excitation as a function of energy has been extracted from LET data over a significant portion of the Brillouin zone. Figure 5 highlights the typical features of phonon ACs: acoustic modes bend and lose linearity well before reaching the zone boundary and they exchange intensity with the optical branch, which tends to drift apart after the AC. ACs are also evident along the Γ–Y direction from INS data in Fig. 4d, but are more complex to unravel. Indeed, more than one low-energy AC occur around the same $q$ along Γ–Y, as confirmed by DFT calculations (see the following section).

**DFT simulations of 4D-INS data.** Preliminary force field calculations derived from DFT results had been used in ref. [30] for a first theoretical evaluation of the phonon dispersion curves of $[VO(acac)_2]$. More accurate results have now been obtained by employing DFT with the finite-displacements method in a supercell. It is worth noting that calculations of phonons for a generic Brillouin zone point by DFT is a daunting task, as the number of atoms that needs to be included is usually very large and approaches the computational limits of this method. Moreover, the accurate description of molecular crystals is particularly challenging due to the known shortcomings of DFT in describing van der Waals (vdW) interactions[51–53]. The latter are expected to be particularly weak in $[VO(acac)_2]$, which is highly volatile as are most β-diketonate neutral complexes.

A periodic $3 \times 3 \times 3$ supercell of $[VO(acac)_2]$ comprising 1620 atoms was used to determine all the translational symmetry-inequivalent reticular force constants. This calculation gives access to vibrational properties for the entire Brillouin zone and is here used to evaluate phonon frequencies and eigenvectors along the three directions of interest. The results reported in Fig. 6 show acoustic modes up to 5 meV and, most importantly, optical branches lying at very low energies and displaying ACs with the acoustic ones, in agreement with the experimental results. Indeed, Fig. 6 also shows the energies of some representative phononic excitations extracted from 1D cuts as in Fig. 2, superimposed to DFT calculations. This comparison shows that calculated phonon energies reproduce important qualitative features of the experimental findings, such as the presence of low-lying optical modes and ACs with acoustic modes. However, a rescaling of 13% has been uniformly applied to all phonon modes, in order to lower

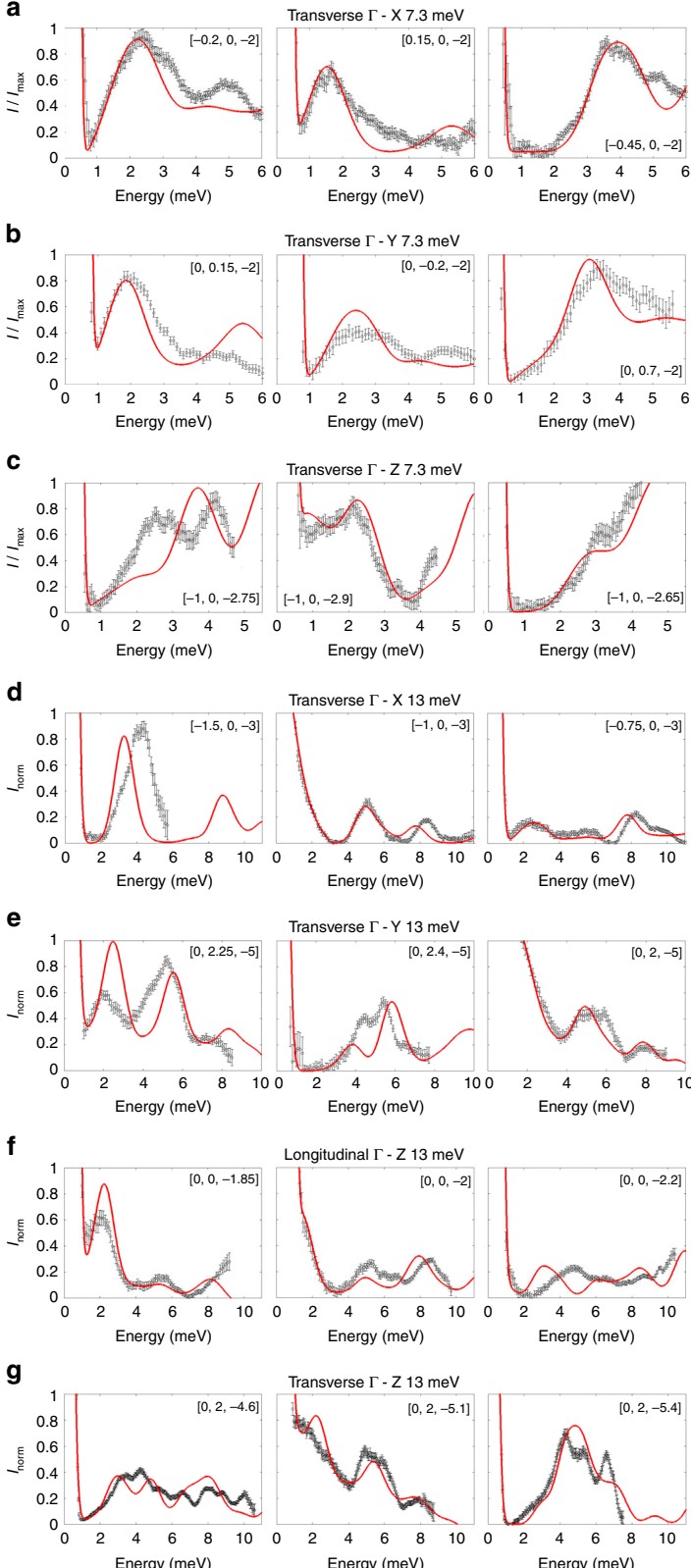

**Fig. 2 Phononic excitations intensities vs energy.** Black scatters: Representative examples of [VO(acac)$_2$] phononic excitations as a function of energy, obtained from LET data with $E_i = 7.3$ meV (**a**–**c**) and $E_i = 13$ meV (**d**–**g**) at $T = 5$ K. The 1D cuts have been obtained by integrating the 4D datasets over the three components of the neutron scattering vector $\mathbf{Q} = [\eta, \zeta, \xi]$ (expressed in terms of reciprocal lattice vector units) around the desired values, with error bars representing the SE. For data obtained with $E_i = 7.3$ meV the intensity is normalised for the maximum in each panel, while for data obtained with $E_i = 13$ meV the intensity is normalised in order to saturate acoustic modes and give prominence to optical ones. Red lines: intensity curves as a function of energy calculated with Eq. (3) (see Methods) and with phonon energies and polarisation vectors obtained from DFT.

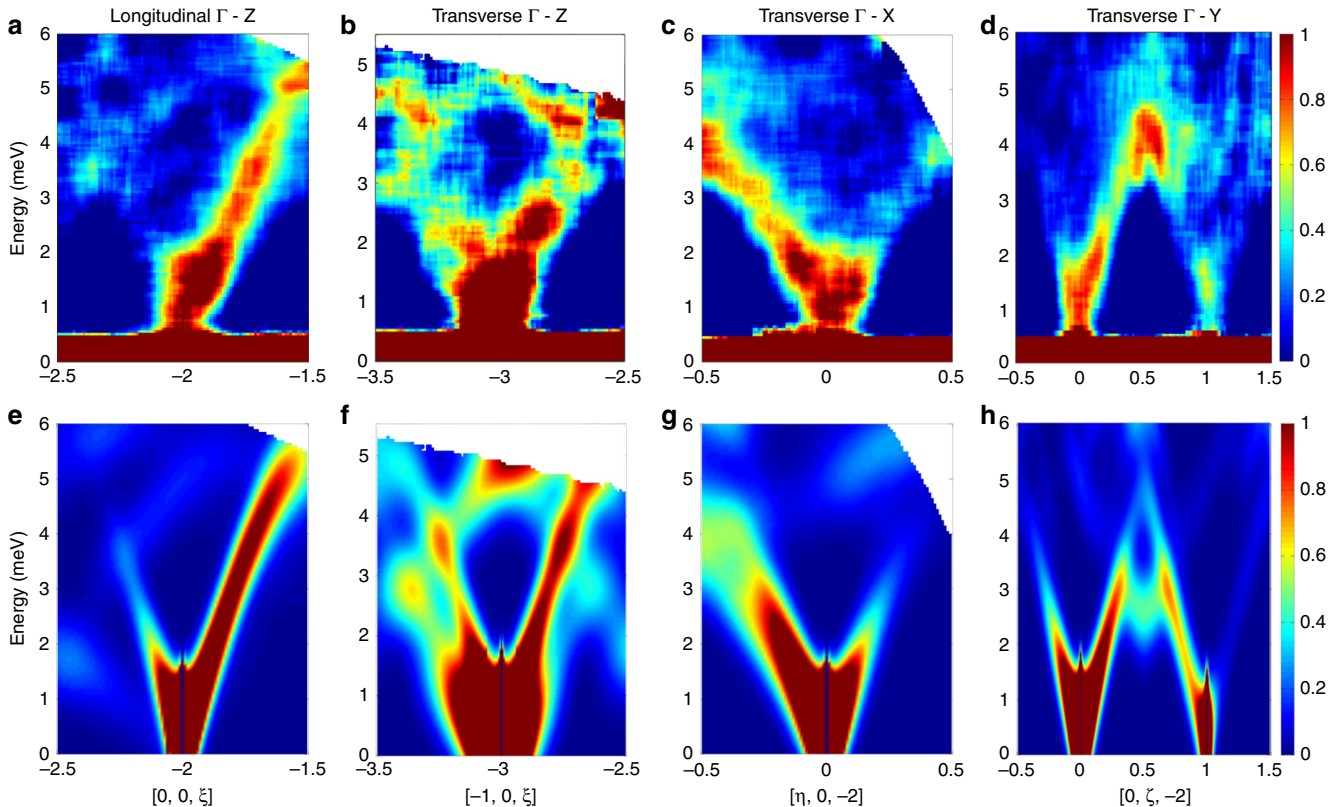

**Fig. 3 Phonon dispersions up to 6 meV.** In **a**–**d** we show representative examples of [VO(acac)$_2$] phonon dispersions measured on LET with $E_i = 7.3$ meV and $T = 5$ K. This incident energy allowed us to resolve acoustic modes up to 6 meV. Dispersions curves have been obtained integrating over two out of the three components of the neutron scattering vector $\mathbf{Q} = [\eta, \zeta, \xi]$ (expressed in terms of reciprocal lattice vector units) around different Bragg peaks. **a** Shows longitudinal acoustic modes along the symmetry direction Γ–Z around [0, 0, −2]. **b**–**d** Report transverse acoustic modes along three principal symmetry directions: Γ–Z around [−1, 0, −3], Γ–X around [0, 0, −2] and Γ–Y around [0, 0, −2] and [0, 1, −2]. The colour bar reports the intensity normalised for the maximum in each panel. Simulated dispersion curves along the same directions are reported in **e**–**h** and were calculated with Eq. (3) (see Methods), with phonon energies and polarisation vectors obtained from DFT.

the calculated phonon energies $\omega_j(\mathbf{q})$ and better reproduce INS data. The observed discrepancies are consistent with other results in literature for molecular crystals[54–56] and are mainly due to the limitations of DFT methods in describing vdW interactions in these systems[51–53], especially at low energies. However, the comparison of phonon frequencies alone is not enough to guarantee the accuracy of DFT simulations. Indeed, it is crucial to verify the normal modes composition, since it enters in the definition of spin-phonon coupling coefficients and deeply affects the calculated spin dynamics. Our experiment probes also this composition, because the INS cross-section is determined by phonon eigenstates.

To investigate this aspect, we have simulated the coherent scattering function $S(\mathbf{Q}, \omega)$ of Eq. (3) (see Methods), starting from phonon energies $\omega_j(\mathbf{q})$ and normal modes polarisation vectors $\boldsymbol{\sigma}_j^d(\mathbf{q})$ calculated by DFT. The simulations of 4D-INS spectra are reported in Fig. 2 (red lines) for intensity vs energy 1D plots and in Figs. 3e–h and 4e–h for 2D colour-maps. Our calculations show an overall qualitative agreement with the observed excitations and dispersions. In particular, it is evident from Figs. 3 and 4 that, while phonon dispersions are symmetric with respect to the Brillouin zone centre, neutron intensity patterns are not, as experimentally observed. However, quantitative discrepancies are evident especially from Fig. 2.

Given that DFT results are able to reproduce important qualitative features of experiments, we can further analyse them in order to extract the atomic motions associated with specific

phonons. Of particular interest is the nature of acoustic and low-lying optical phonons near the AC points. For instance, Fig. 7 shows atomic displacements associated to a transverse acoustic mode before and at the AC with a low-lying optical branch along the Γ–X direction. It is evident that, while before the AC the displacements have a translational character (Fig. 7a), at the AC molecular vibrations are profoundly changed and the rigid translation of the molecule typical of acoustic modes is now rather small. Indeed, Fig. 7b shows that the predominant motions induced by the acoustic phonon at the AC are a rigid translation strongly mixed with bending and torsions of acetylacetonate ligands and with methyl groups rotations, causing the VO$^{2+}$ moiety to move with respect to the basal plane. Thus, both distances and angles between the V centre and single-bonded oxygen ions are modified by this mixed acoustic mode, while, as expected, the double bond of the VO$^{2+}$ moiety remains rigid. We have also verified that these distortions are the same induced by the optical phonon, thus demonstrating a strong admixing of the two modes at the AC. Similar conclusions can be drawn for the AC involving the transverse acoustic mode and the low-lying optical branch along the Γ–Z direction.

## Discussion
Having direct access to phonon dispersions, and in particular for the first time to ACs between acoustic and optical modes, it is now interesting to focus on their effect on the relaxation dynamics. Indeed, the here-calculated phonons energies and eigenvectors enable the evaluation of the relaxation dynamics in

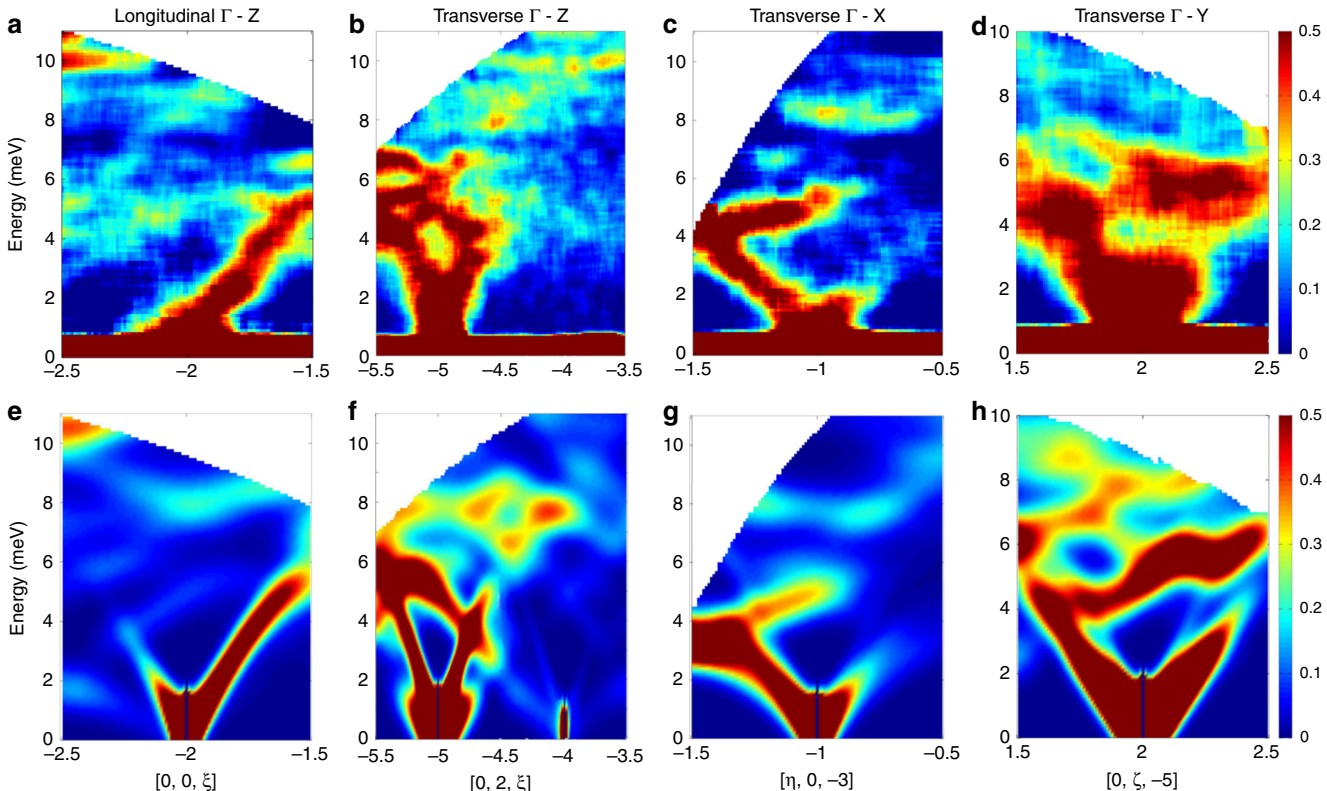

**Fig. 4 Phonon dispersions up to 11 meV.** In **a–d** we show representative examples of [VO(acac)$_2$] phonon dispersions measured on LET with $E_i = 13$ meV and $T = 5$ K. Dispersions curves have been obtained integrating over two out of the three components of the neutron scattering vector $\mathbf{Q} = [\eta, \zeta, \xi]$ (expressed in terms of reciprocal lattice vector units) around different Bragg peaks. **a** Shows longitudinal acoustic modes along the symmetry direction Γ–Z around [0, 0, −2], as in Fig. 3a. With this incident energy we have been able to resolve not only the acoustic but also optical phonon modes up to 11 meV. **b–d** Display transverse modes along three principal symmetry directions: Γ–Z around [0, 2, −5] and [0, 2, −4], Γ–X around [−1, 0, −3] and Γ–Y around [0, 2, −5]. The colour bar reports the intensity normalised for the maximum in each panel. Simulated dispersion curves along the same directions are reported in **e–h** and were calculated with Eq. (3) (see Methods), with phonon energies and polarisation vectors obtained from DFT.

this prototype molecular qubit. A first ab initio simulation of direct relaxation processes has just been performed[57]. In that contribution the authors showed that in high external magnetic field the dominant mechanism of spin relaxation comes from the modulation of the **g** tensor of the Spin Hamiltonian Zeeman term by low-energy acoustic modes. Starting from those findings, we here investigate the effect of ACs on this mechanism by calculating for the first time the $q$-resolved spin-phonon coupling coefficients. In particular, we report the squared norm of each coefficient for each phonon mode $j$ as a function of the phonon wave vector **q**:

$$V^2_{\mathrm{SPh}}(\omega_j, \mathbf{q}) = \sum_{\alpha,\beta=x,y,x} \left( \frac{\partial g_{\alpha,\beta}}{\partial Q_{j,\mathbf{q}}} \right)^2, \quad (1)$$

where $g_{\alpha,\beta}$ are the components of the **g** tensor and $Q_{j,\mathbf{q}}$ are those of the normal modes[57]. Results for the AC along Γ–X are reported in Fig. 8 (see also Supplementary Fig. 2 for results along Γ–Z) and demonstrate that, in general, optical modes are characterised by stronger spin-phonon couplings coefficients with respect to the acoustic ones. However, at the ACs the stronger spin-phonon coupling of the almost flat optical modes is partially transferred to the dispersive acoustic modes, which display a maximum of $V^2_{\mathrm{SPh}}$. This is evident for both the AC involving the transverse acoustic mode analysed in Figs. 5 and 7 and for the AC involving the longitudinal one at higher energies. Figure 8 shows that the mixing of the modes is not negligible even well before the AC at lower |**q**| values and lower energies, where the acoustic mode is still strongly

dispersive. Indeed, the enhancement of spin-phonon coupling for the nominally acoustic mode starts at energies significantly smaller than the AC one. It is well known that low-energy dispersive modes are crucial for magnetic relaxation. Hence, the small optical component present in the nominally acoustic mode at low energies can significantly influence magnetic relaxation, by modulating the **g** tensor with larger probability and in a wider energy range than without the mixing of modes.

These results about the effect of ACs on the spin-phonon couplings further confirm the importance of a complete description of phonons to understand relaxation dynamics in MNMs[18]. Recently, ab initio calculations have been used to predict the contribution of individual vibrational modes to the spin-phonon coupling[23,31,58]. The agreement with experimental results was limited because only vibrations at the Γ-point were included[31] or due to the gas-phase approximation[5]. For comparison, gas-phase calculations of the magneto-vibrational couplings for the present molecule (see Methods) have been performed and are reported in Supplementary Note 1. At last, it was shown in thermoelectric materials with low-energy optical branches[48–50] that the strong mixing of the two modes involved in the ACs decreases acoustic phonons group velocity and also phonon lifetimes. Indeed, the presence of low-lying optical modes is also expected to enhance phonon–phonon scattering processes. This decrease of phonon lifetimes could further affect magnetic relaxation[23].

The scenario evidenced by this study is expected to be common to other MNMs. Indeed, the presence of soft coordination bonds

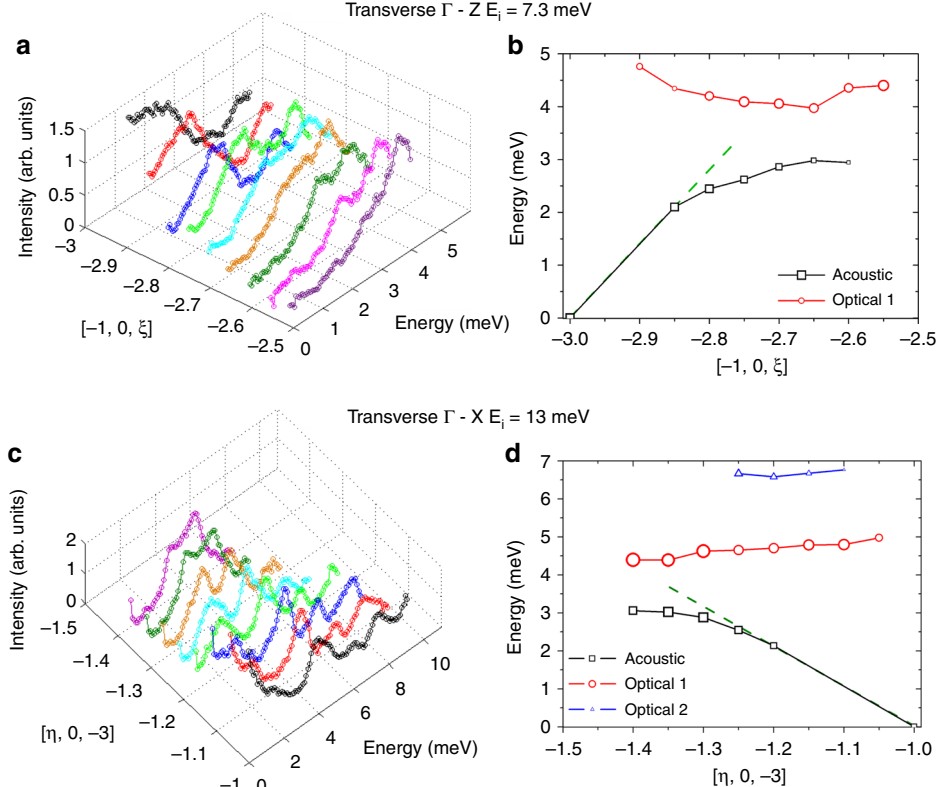

**Fig. 5 Phonon anti-crossings.** Panels **a** and **c** report 1D cuts as a function of energy of the LET data, obtained integrating around different values of **Q** for **a** Γ–Z around [−1, 0, −3] with $E_i$ = 7.3 meV (Fig. 3b) and **c** Γ–X around [−1, 0, −3] with $E_i$ = 13 meV (Fig. 4c). Panels **b** and **d** report the position of the peaks obtained from the 1D cuts of the data for both acoustic (black scatters) and optical modes (red and blue scatters). The intensity of each peak is translated into scatter dimension and the dashed green lines interpolate the linear behaviour of the acoustic modes. The data show the typical features of phonon ACs: bending of the acoustic mode, intensity exchange between acoustic and optical branches, with the latter drifting apart after the AC.

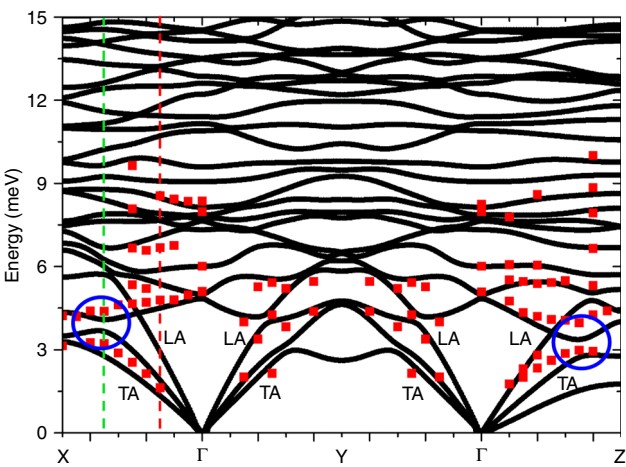

**Fig. 6 DFT calculations of phonon dispersions compared to experimental excitation energies.** DFT calculations of the low-energy phonon dispersion curves of $[VO(acac)_2]$ up to 15 meV along the path $X(0.5, 0, 0)$–$Γ(0, 0, 0)$–$Y(0, 0.5, 0)$–$Γ(0, 0, 0)$–$Z(0, 0, 0.5)$ (black lines), where the coordinates are expressed in the reciprocal lattice basis units. Red scatters are the energies of experimental excitations extracted by fitting 1D cuts of the data as in Fig. 2 (error bars as SD are of the order of scatters dimension). Not all the modes are visible in the considered configurations. Blue circles highlight ACs between acoustic and low-lying optical branches along Γ–X and Γ–Z. Red and green dashed lines indicate the **q** vectors at which we have investigated the atomic displacements far from and at the ACs, respectively.

and vdW interactions strongly reduces the energy of rotational and intra-molecular vibrations, leading to a non-negligible admixing between acoustic and optical branches. This effect is already visible in crystals of small-size molecules such as arenes[54,55]. Furthermore, being able to correlate the molecular structure of these systems with their experimentally tested vibrational and phonon spectra, will also enable the development of new strategies for the design of new and optimised molecular qubits and bits. Our results point out that long coherence times or slow magnetic relaxation are undermined by the presence of acoustic-optical phonons ACs. Thus, future synthetic strategies should, for instance, reduce the presence of soft coordination bonds with the aim to shift optical branches to higher energies and thus remove some of the ACs. Few ACs could be also eliminated by increasing the crystal symmetry. However, this would only affect crossings along symmetry directions, while ACs would still be present along generic **q** directions.

At last, the present study represents an important test for vdW-corrected DFT in describing phonon dispersions. We showed that while qualitative features of experimental results are reproduced, a full quantitative reproduction of both eigenvalues and eigenvectors will require further improvements of DFT to treat vdW interactions. The DFT community involved in this field commonly use molecular crystals' cohesive energy[59,60] as benchmark test, whose direct comparison with experimental estimation is known to represent a challenge, due to the presence of multiple effects contributing to the measured values (e.g., finite-temperature and zero-point-energy effects[61]). Conversely, here we have a direct comparison between the practically zero-

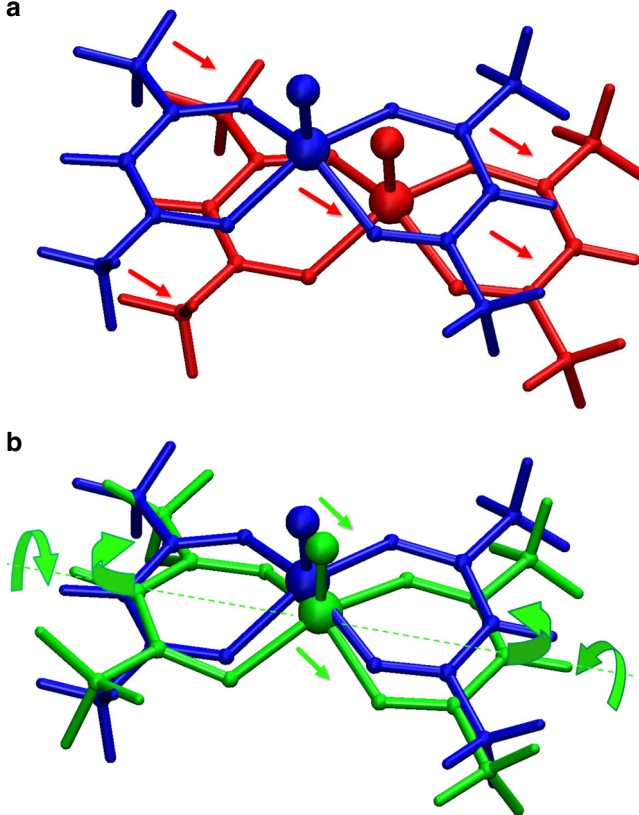

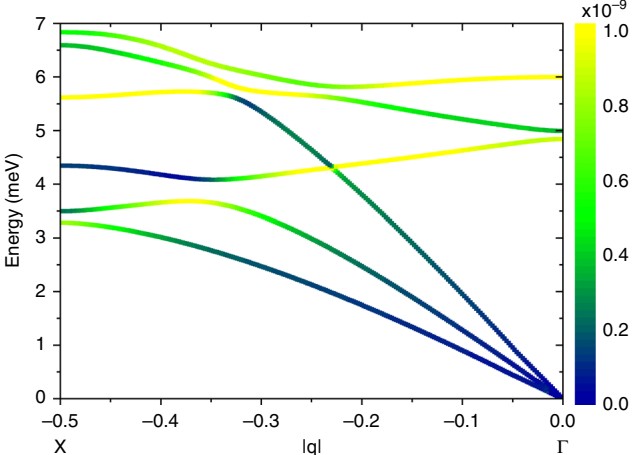

**Fig. 8 DFT spin-phonon couplings.** Colours map spin-phonon couplings squared norm $V^2_{SPh}(\omega_j, \mathbf{q})$ calculated as in Eq. (1) for the low-energy phonon modes along $\Gamma$–$X$, as a function of $|\mathbf{q}|$. At the ACs the strong coupling of the optical modes is transferred to the acoustic ones.

**Fig. 7 DFT atomic displacements. a** The DFT equilibrium molecular structure (blue ball-and-stick) is superimposed with the distorted one according to the nature of the second transverse acoustic mode along $\Gamma$–$X$ at $\mathbf{q} = (0.145, 0.0, 0.0)$ (red ball-and-stick, red dashed line in Fig. 6). **b** The DFT equilibrium molecular structure (blue ball-and-stick) is overlapped with the distorted one according to the nature of the second acoustic mode along $\Gamma$–$X$ at $\mathbf{q} = (0.345, 0.0, 0.0)$ (green ball-and-stick, green dashed line in Fig. 6). Arrows sketch the small rigid translation of the molecule which is now strongly mixed with the bending of the acetylacetonate ligands and their torsion around the dashed axis.

temperature experimental data and the calculated phonon energies, dispersions and polarisation vectors. This comparison for molecular crystals of such a complexity is almost unique and it could represent also a new standard in the benchmarking of vdW correction schemes.

## Methods

**Synthesis and crystal structure of the samples.** [VO(acac)$_2$] is a prototypical molecular qubit containing a single magnetic ion: We use the term molecular nanomagnets (MNMs) also for this new generation of molecular bits and qubits.

[VO(acac-d$_7$)$_2$] was obtained by reaction of a basified (NaOD 99% D) deuterated water solution (D$_2$O 99.8% D) of acetylacetone-d$_8$ (98% D) and VOSO$_4$. The precipitate of [VO(acac-d$_7$)$_2$] was filtered and recrystallised in inert atmosphere with deuterated acetone-d$_6$ (99.95% D). Single crystals were obtained by slow evaporation of the solvent (2 weeks). The effective percentage of deuteration of [VO(acac-d$_7$)$_2$] was estimated by electron spray ionisation mass spectrometry, and it was found to be 98 ± 1%. All crystals were indexed using a single-crystal X-ray diffractometer. The investigated sample included about 40 single crystals for a total mass of about 1 g, co-aligned one-by-one using the ALF crystal alignment facility at ISIS. Since [VO(acac)$_2$] crystallises in the triclinic centrosymmetric space group, this procedure represented an additional challenge for the experiment, but the final mosaicity of the sample was enough to obtain sufficiently focused Bragg peaks (see also Supplementary Fig. 1).

**4D-INS experiment.** The 4D-INS experiments on [VO(acac)$_2$] were performed on the LET cold neutrons time-of-flight spectrometer at the ISIS Neutron and Muon source[34]. LET is a direct geometry spectrometer with $\pi$-steradians position-sensitive detectors bank divided in about $10^5$ pixels, covering 180° of azimuthal angle and ±30° out-of-plane.

Forty [VO(acac)$_2$] single crystals were placed on five different aluminium plates using a fluorinated oil, then stacked on a custom-made sample holder. Having aligned the crystals with the reciprocal vector $\mathbf{a}^*$ perpendicular to the scattering plane, we were able to explore large portions of the [VO(acac)$_2$] reciprocal space in the $\Gamma$–$Z$ and $\Gamma$–$Y$ directions and a smaller portion in the $\Gamma$–$X$ one (see Supplementary Fig. 1). Measurements were performed by rotating the crystals (in steps of 1°) about the vertical axis.

Two incident neutron energies of 7.3 and 13 meV were selected in repetition rate multiplication mode, with an energy resolution (full-width at half-maximum) of 0.23 and 0.55 meV at the elastic line, respectively. All measurements were performed at $T = 5$ K.

**DFT calculations.** All the structural optimisation and Hessian calculations were performed with the CP2K software[62] at the level of DFT with the Perdew–Burke–Ernzerhof functional[63] including Grimme's D3 vdW corrections[64,65]. Indeed, the requirement of vdW-corrected functionals for the modelling of molecular crystals is by now confirmed and settled by several results in literature[51–53,55]. A double-zeta polarised (DZVP) MOLOPT basis set and a 600 Ry of plane-wave cutoff were used for all the atomic species. All the translational symmetry independent force constants were computed by finite difference approach with a 0.01 Å atomic displacements.

The optimisation of the isolated molecule has been done with a plane-wave cutoff of 850 Ry and by turning off the periodic boundary conditions. An integration step of 0.01 Å was used for the calculation of the isolated molecules force constants.

The coefficients $\left(\partial g_{\alpha\beta}/\partial Q_{j\mathbf{q}}\right)$ were calculated as

$$\left(\frac{\partial g_{\alpha\beta}}{\partial Q_{j\mathbf{q}}}\right) = \sum_{l}^{N_{cells}} \sum_{is}^{N,3} \sqrt{\frac{\hbar}{N_q \omega_{j\mathbf{q}} m_i}} e^{i\mathbf{q}\cdot\mathbf{R}_l} L_{is}^{j\mathbf{q}} \left(\frac{\partial H_s}{\partial X_{is}^l}\right),\qquad(2)$$

where $X_{is}^l$ is the $s$ Cartesian coordinate of the $i$th atom of $N$ with mass $m_i$, inside the unit-cell replica at position $\mathbf{R}_l$ and $N_q$ is the number of $q$-points used. Finally, $\omega_{j\mathbf{q}}$ and $L_{is}^{j\mathbf{q}}$ are the $j$th eigenvalue and eigenvector of the Hessian matrix at the $\mathbf{q}$ point. The Cartesian derivatives $\left(\partial g_{\alpha\beta}/\partial X_{is}\right)$ were obtained from ref. [57].

**Data analysis and simulations.** The 4D-INS technique is named after the possibility to measure the four-dimensional scattering function $S(\mathbf{Q}, \omega)$, i.e. as a function of the transferred energy and of the three components of the transferred momentum $\mathbf{Q}$. The one-phonon coherent scattering function $S(\mathbf{Q}, \omega)$ is defined as[66]

$$\begin{aligned}S(\mathbf{Q}, \omega) \propto \sum_{j,\mathbf{G},\mathbf{q}} &\left|F_{j,\mathbf{q}}(\mathbf{Q})\right|^2 \frac{1}{2\omega_j(\mathbf{q})} \Big[n_j(\mathbf{q})\delta(\omega + \omega_j(\mathbf{q}))\delta(\mathbf{Q} + \mathbf{q} - \mathbf{G}) \\ &+ (n_j(\mathbf{q}) + 1)\delta(\omega - \omega_j(\mathbf{q}))\delta(\mathbf{Q} - \mathbf{q} - \mathbf{G})\Big],\end{aligned}\qquad(3)$$

where $\omega_j(\mathbf{q})$ are the phonons' frequencies. The summation in Eq. (3) is extended to all phononic branches $j$. The momentum conservation law of the scattering events involves the neutron scattering vector $\mathbf{Q}$, the reciprocal lattice vector $\mathbf{G}$ and the

phonon wave vector $\mathbf{q}$, while the energy conservation depends on the neutron energy transfer $\omega$. The term $n_j(\mathbf{q}) = (\exp(\beta\omega_j(\mathbf{q})) - 1)^{-1}$ is the phonon Bose factor with $\beta = (k_B T)^{-1}$.

The structure factor $F_{j,\mathbf{q}}(\mathbf{Q})$ takes into account interference effects between the different atoms $d$ in the unit cell:

$$F_{j,\mathbf{q}}(\mathbf{Q}) = \sum_d \frac{b_d}{\sqrt{m_d}} e^{-W_d(\mathbf{Q})} e^{i\mathbf{Q}\cdot\mathbf{R}_d}(\mathbf{Q}\cdot\boldsymbol{\sigma}_j^d(\mathbf{q})), \quad (4)$$

where $e^{-W_d(\mathbf{Q})}$ is the Debye-Waller factor, $b_d$ is the coherent scattering length, $m_d$ the mass and $\mathbf{R}_d$ the position vector in the real space of each atom $d$ and $\boldsymbol{\sigma}_j^d(\mathbf{q})$ are the phonons' polarisation vectors.

Time-of-flight data reduction were performed with Mantid analysis suite[67] and measurements for different rotation angles were combined using HORACE[46], yielding four-dimensional $S(\mathbf{Q},\omega)$ datasets for the selected incident energies. HORACE also allows to slice these 4D datasets into 2D slices along specified trajectories in the $(\mathbf{Q}, E)$-space, in order to visualise phonon dispersion along different directions, and into 1D curves, to visualise excitations intensities as a function of energy around desired $\mathbf{Q}$ values. In order to reduce the noise-to-signal ratio, data along each symmetry direction were integrated over the other two within a $\pm 0.1$ range (in reciprocal lattice vector units) centred around the Bragg peak (for longitudinal $\Gamma$–$Z$ and transverse $\Gamma$–$X$ and $\Gamma$–$Y$ modes with $E_i = 7.3$ meV it is necessary to integrate within a $\pm 0.2$ range). Empty sample holder data were subtracted, corrections for multiple-scattering flat backgrounds and a 7 bins smoothing were also applied. Low-$Q$ regions of the reciprocal space were selected for our in-depth analysis, in order to better focus on coherent scattering. 1D cuts as a function of energy were obtained by integrating 2D slices over the surviving direction in a $\pm 0.1$ interval centred around the desired $\mathbf{Q}$ value and by applying a seven bins smoothing, with error bars representing the SE. They were then fitted with Gaussian functions in order to extract excitation energies in Fig. 6.

Data simulations were performed by calculating the scattering cross-section in Eq. (3) with DFT calculated phonon energies $\omega_j(\mathbf{q})$ and normal modes polarisation vectors $\boldsymbol{\sigma}_j^d(\mathbf{q})$. To reproduce INS data, a small rescaling of about 13% was uniformly applied to all the calculated phonon energies along all the symmetry directions. We then assumed a Gaussian line-shape with $\sigma = 0.6$ meV. For the simulation of 1D cuts in Fig. 2 we also added the contribution of the elastic signal.

## Data availability

Raw data from the INS experiment were generated at ISIS Neutron and Muon Source and are available at doi.org/10.5286/ISIS.E.86390979 and doi.org/10.5286/ISIS.E.86390999. Derived data supporting the findings of this study are available from the corresponding authors upon reasonable request.

## Code availability

The custom Matlab and Fortran codes for data analysis and simulations are available from the corresponding authors upon reasonable request.

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

## Acknowledgements

Financial support from PRIN Project 2015 No. HYFSRT of the Italian MIUR, from the European project QuantERA 2017–SUMO and Science Foundation Ireland (grant 14/IA/ 2624) are gratefully acknowledged. E.G. acknowledges the support of "Fondazione Angelo della Riccia" and of the PRISM Project of the call "FIL-Quota incentivante 2019" of the University of Parma. Computational resources were provided by the Trinity Centre for High-Performance Computing (TCHPC) and the Irish Centre for High-End Computing (ICHEC). The authors also acknowledge the CRIST laboratory at University of Florence for the help and support in crystals indexing. Experiments at the ISIS Neutron and Muon Source were supported by a beamtime allocation RB1710499 from the Science and Technology Facilities Council. Dr. Duc Le is also gratefully acknowledged for his assistance on phonon data analysis.

## Author contributions

E.G., L.T., D.J.V. and T.G performed the neutron scattering experiment on crystals synthesised by M.A. and aligned by L.T. and R.S. Data treatment was made by E.G., D.J.V., T.G. and S.C., while A.L. and S.S. performed DFT calculations. Data analysis and simulations were made by E.G., D.V., P.S. and S.C. P.S., T.G., R.S. and S.C. developed the idea behind this work. E.G. and S.C. wrote the manuscript with inputs from all coauthors.

## Competing interests

The authors declare no competing interests.
