## [Peer Review File · Nature Communications]

Reviewers' comments:

Reviewer #1 (Remarks to the Author):

The authors have presented an elegant demonstration of the power of modern neutron scattering instrumentation coupled to computational methods to extract detailed phonon dispersions in single-hit experiments rather than the step-wise cuts more traditionally associated with triple-axis spectrometry (TAS). This approach - termed here 4D-INS - is now readily accessible on time-of-flight (as used here) and flat cone TAS instruments. The sample of this study is a small molecule having a spin moment and as such the phonon bath is of interest as it may modulate the spin relaxation behaviour. This is a problem of some significance and the manuscript is worthy of publication; I must say that the visual impact of the measured vs. calculated responses in figure 2 & 3 is striking.

I do have some issues with the manuscript, however I do not anticipate them to be major obstacles to publication. The authors should be able to readily address them.

Firstly there are a few grammatical flaws in the text.

I have let many go, but the line '...volatile as most β -diketonate....' needs a verb - I suggest placing 'are' between 'as' & 'most'.

Then there needs to be an article such as 'the' in the line '...., also in presence of very....', as in '...in the presence of...'.
'Occurs' is probably a better way of expressing 'take place' in the text below FIG. 2.

The discussion section is a weak point of the manuscript - it tends to repeat facets already discussed and does not really add much to the manuscript. In my opinion the 'Discussion' section of a paper should be just this and is the section in which more detailed understanding is developed; whilst the authors in part achieve this, it really needs a rewrite to be more focused.

Much of the 'Methods' section is written in an odd style - why the repeated use of 'have/has been' rather than the more direct reportage of 'was', 'were', etc? Comes across as a bit clunky.

The authors mention a 13% down-scaling of the calculated phonon modes in a rather casual manner. Is this a uniform scaling across all modes and energies? How does this correspond to similar studies - there have been many reports of lattice and phonon modes using DFT & neutron scattering (maybe not with the full 4D dispersion as many such measurements have been reported on polycrystalline samples), but all the same, scaling behaviours would be similar. The authors should make some reference to this and discuss their 13% relative to any scalings used elsewhere - some may consider a >10% scaling to be a significant error. I tend to agree with the authors that such agreement is reasonable given the poor treatment of intermolecular interactions in DFT, but their case for using such a scaling can be strengthened by providing contemporary examples of other work.

Reviewer #2 (Remarks to the Author):

This manuscript discusses results of the neutron scattering measurements of VO(acac)₂. These results may be of interest to the part of the neutron scattering community specializing in phonons. This study is solid, and the results/interpretations are valid. However, I don't see any significant broader impact for a number of reasons.

1. The main finding of anti crossings between acoustic and optic phonons in a crystal prepared from these molecules can only happen in crystals where acoustic branches exist. It cannot occur in single molecules where there are no acoustic phonons. So the main result of this work is completely irrelevant to single-molecule cubits.

2. The anticrossings reported here are generic and occur almost all the time, so the novelty of this work is not that high.

3. It would be more interesting to explore higher energy phonons, which are closer to what would be observed in actual qubits, but this has not been done due to limitations of the LET spectrometer.

4. As a relatively minor point, the agreement with the DFT calculations is not as good as claimed, since the phonons near the X-point are about 20% higher in energy than calculated, and the experimental anti crossing gap is much higher than the theoretical one.

This article would be more suited for a specialized journal than for Nature Communications.

Reviewer #3 (Remarks to the Author):

The manuscript details the application of a new technique to studying phonons in a molecular qubit system. Vibrations, of molecules or their local matrices, are an intense focus in the world of molecular magnetism. In particular, understanding how vibrations correlate to magnetic relaxation time is a notable step toward the holy grail of finding long relaxation times at high temperatures. The manuscript by Garlatti and coworkers takes an important step toward that goal by using a new technique – 4D inelastic neutron scattering. They apply this technique for the first time to a molecular system, revealing evidence of phonon anticrossings between acoustical and optical branches.

The application of the new technique to identifying phonons and how they specifically perturb a metal complex is significant and warranting publication. However, there are two aspects of the work that I think either require heavy revision or publication elsewhere. First, the manuscript is written in a way that renders it very difficult to access by those who could potentially use the work to design new molecular qubits by vibration tuning. I'd recommend some revision to help the synthetic folks out a little bit, especially if this is going into such a high-impact journal.

Second, the tie to magnetism (the most important implication) is stated in a manner that is too vague to make this an obvious high-impact publication. How exactly will the avoided crossings impact magnetism? Is it through controlling phonon lifetimes? How would the lifetime of a phonon or molecular vibration be expected to impact the spin relaxation times? There are a lot of manuscripts coming out attempting to tie vibrations to magnetism, and the foregoing questions are probably much too big to answer in one manuscript (and the given data do not answer all of these questions), but some additional development is necessary in my opinion. Without better explanation and context from the rest of literature (including a recent paper by Moseley et al. *Eur. J. Inorg. Chem.* 2019, 8, 1055), the impact of the manuscript seems inflated with no clearly articulated link between what was discovered/measured and an impact on magnetism.

Reviewer #4 (Remarks to the Author):

The paper provide convincing results on the interest of 4D-INS techniques coupled with DFT calculations to address the question of the impact of phonon on the relaxation processes in MNMs, and particularly the role of the AC between optical and acoustical branches. The experimental and theoretical parts are well presented and clear even for non specialist, the text is very clear. Then, it is highly relevant to publish it in Nat. Comm.

About the impact of vdW interactions inducing "admixture between acoustic and optical branches", I am wondering if DFT calculations using the same functional but without D3 corrections could be

performed. Indeed, vdW interactions should be killed in these calculations and the comparison with the results presented in the paper could give more weight to the point above.

I suggest some simplification of the text:

* "state-of-the-art DFT" appears 5 times in the text. I am convinced that the calculations have correctly been done and that the results are good enough to support the conclusions of the authors. But such repetition does not provide more proof and sounds like if authors were "self-satisfied" (I hope the meaning in English is clear, I am not sure of the translation, my aim is not to offend the authors).

*some sentences/ideas appears twice without, the text should then be lighten:

- page 3 "the dot product"..."allows to select between longitudinal and transverse phonon modes"

/ "An important feature of INS technique is the possibility to experimentally distinguish between longitudinal and transverse modes by properly selecting the Q and phonon ..."

- the coupling between optical and acoustic branches and its impact is clearly presented, but I feel that authors repeat a bit too much this feature, as if they were not sure that it is clear to the reader.

We would like to **thank all the Referees**, because their comments allowed us to significantly strengthen our paper with new calculations and an improved discussion and presentation.

We thank **Referee 1, 3 and 4 for recognizing the importance of our results** and Referee 1 and 4 for recommending publication in Nature Communications. Referee 3 asks to improve readability for a broad audience and to do an “additional development” towards addressing how “will the avoided crossings impact magnetism”. Following the remark of Referee 3, we have performed **new detailed calculations** showing that **avoided crossings between acoustical and optical phonons have crucial effects** on spin-phonon couplings, and hence on magnetic relaxation. These new results confirm the importance of a complete description of phonons to understand relaxation dynamics in Molecular Nanomagnets. We have also rewritten the discussion and significantly modified several parts of the paper to **improve the clarity of the presentation** and to better underline the consequences of our results.

At last, Referee 2 finds “This study is solid, and the results/interpretations are valid”, but is not convinced of the broad interest of our results, mainly because phonons do not exist in single isolated molecules. However, the interest of our work is not limited to molecular qubits. Indeed, **magnetic relaxation in crystals** of molecular nanomagnets (MNMs) has been and is the **pivotal point of most of the research** in molecular magnetism. However, phonons, the main actors causing relaxation, have never been directly and deeply experimentally studied in MNMs before. Hence, the present results are of great relevance for the whole molecular magnetism community. Even if we limit to quantum information processing applications, the **vast majority of the studies on molecule-based quantum computing are not on individual molecules**. We believe that the judgment of Reviewer 2 could have been affected by some weaknesses in the presentation of our results and their consequences in the previous version of the manuscript.

Please find below our detailed answers to Reviewers’ remarks and the corresponding modifications of the manuscript. A full list of changes is reported at the end.

Reviewer #1:

“The authors have presented an elegant demonstration of the power of modern neutron scattering instrumentation coupled to computational methods to extract detailed phonon dispersions in single-hit experiments rather than the step-wise cuts more traditionally associated with triple-axis spectrometry (TAS). This approach - termed here 4D-INS - is now readily accessible on time-of-flight (as used here) and flat cone TAS instruments. The sample of this study is a small molecule having a spin moment and as such the phonon bath is of interest as it may modulate the spin relaxation behaviour. This is a problem of some significance and the manuscript is worthy of publication; I must say that the visual impact of the measured vs. calculated responses in figure 2 & 3 is striking.”

We thank the Reviewer for the positive comments about the significance and the results of our manuscript. We also thank them for their very useful remarks and suggestions. Please find below point-by-point answers and subsequent changes and additions in the text.

“Firstly there are a few grammatical flaws in the text.

I have let many go, but the line '....volatile as most β -diketonate....' needs a verb - I suggest placing 'are'

between 'as' & 'most'.

Then there needs to be an article such as 'the' in the line '...., also in presence of very....', as in '....in the presence of....'.

'Occurs' is probably a better way of expressing 'take place' in the text below FIG. 2."

We thank the Reviewer for highlighting these inaccuracies that we fully addressed.

"The discussion section is a weak point of the manuscript - it tends to repeat facets already discussed and does not really add much to the manuscript. In my opinion the 'Discussion' section of a paper should be just this and is the section in which more detailed understanding is developed; whilst the authors in part achieve this, it really needs a rewrite to be more focused."

We are grateful to the Reviewer since it helped us to focus on an important weakness in the presentation of our work. Now we reduced the repetitions, also by shortening the summary of the results in the "Introduction" and at the beginning of the "Discussion" section.

Furthermore, we added significant new results to the "Discussion" section. In particular, to have a deeper understanding of the consequences of our findings on magnetic relaxation, we performed additional calculations focused on the effects of Anti Crossings (ACs) in $[\text{VO}(\text{acac})_2]$. We followed the method developed by A. Lunghi and S. Sanvito, arXiv:1903.01424 (2019) to calculate the spin-phonon coupling coefficients for low-energies phonon modes as a function of the phonon wavevector \mathbf{q} (see new equation 1 in the main text). These new results are reported in the new Fig.7 and Fig.S2. We found that at the ACs the significantly stronger spin-phonon coupling of the flat optical modes is partially "transferred" to the dispersive acoustic modes, which respectively display a minimum or a maximum of the spin-phonon coupling. This effect enhances the spin-phonon coupling of acoustic modes not only at the AC, but also at smaller energies and \mathbf{q} , where they're still strongly dispersive. Hence, spin-phonon interactions involving acoustic modes are enhanced in a significant energy range because of this mixing, thus fastening the associated relaxation processes.

"Much of the 'Methods' section is written in an odd style - why the repeated use of 'have/has been' rather than the more direct reportage of 'was', 'were', etc? Comes across as a bit clunky."

We followed the Reviewer suggestion, which improved the readability of the Methods section.

"The authors mention a 13% down-scaling of the calculated phonon modes in a rather casual manner. Is this a uniform scaling across all modes and energies? How does this correspond to similar studies - there have been many reports of lattice and phonon modes using DFT & neutron scattering (maybe not with the full 4D dispersion as many such measurements have been reported on polycrystalline samples), but all the same, scaling behaviours would be similar. The authors should make some reference to this and discuss their 13% relative to any scalings used elsewhere - some may consider a >10% scaling to be a significant error. I tend to agree with the authors that such agreement is reasonable given the poor treatment of intermolecular interactions in DFT, but their case for using such a scaling can be strengthened by providing contemporary examples of other work."

Following the Reviewer suggestion, we added in the paper some comments and references (49-51, 54) on the 13% down-scaling of phonon energies both in the “Results” section on “DFT Simulations of 4D-INS data”.

The applied rescaling is uniform over all the phonon modes and along all the symmetry directions. As we stated at the end of the “Discussion” section, our work represents one of the few examples on a molecular crystal where phonon energies, dispersions and polarization vectors calculated with vdW-inclusive DFT are directly compared with experiments, whereas the usual benchmark for DFT and vdW corrections are lattice parameters and cohesive energies [A. M. Reilly and A. Tkatchenko, *J. Chem. Phys.* 139, 024705 (2013); J. Moellmann and S. Grimme, *J. Phys. Chem. C* 118, 7615 (2014)]. In particular, there are no similar works on Molecular Nanomagnets yielding the same level of description of phonon modes to compare our results with. Nonetheless, as suggested by the Reviewer, we compared our findings with other works on polycrystalline samples or triple-axis spectrometer data. The 13% down-scaling applied to our DFT calculations falls within the range of discrepancies between phonon energies calculated with vdW-corrected methods and INS data found in literature for molecular crystals. The reported discrepancies go from those presented by F. Brown-Altvater *et al.*, *Phys. Rev. B* 93, 195206 (2016) for naphthalene, where an 8% discrepancy is found for intermolecular modes, to the results shown by S. E. Stavretis *et al.* *Eur. J. Inorg. Chem.*, 8, 1119–1127 (2019) for Co^{II} Single-Molecule Magnets, where the INS low-energy spectrum as a function of energy is not well reproduced by vdW-corrected DFT calculations.

It is also worth to stress that the description of properties like phonon modes by vdW-inclusive DFT methods strongly depends on the accuracy of the applied models in every step of the calculation (e.g. calculation of lattice parameters and energies) [G. J. Kearley, M. R. Johnson, and J. Tomkinson, *J. Chem. Phys.* 124, 044514 (2006). J. Maurer *et al.*, *Annu. Rev. Mater. Res.* 49, 1–30 (2019); J. Hoja *et al.*, *WIREs Comput. Mol. Sci.* 7, 1294 (2016)]. Furthermore, among molecular crystals there is a huge diversity of molecular and crystal structures and covalent and non-covalent interactions, all of which impact properties like phonons [J. Maurer *et al.*, *Annu. Rev. Mater. Res.* 49, 1–30 (2019)]. For this reason, several vdW functionals have been proposed during the last few years, making not always clear which vdW correction should be used [F. Tran *et al.*, *Phys. Rev. Mat.* 3, 063602 (2019)]. Indeed, the choice of the correct DFT method for molecular crystals simulations is, in general, still a debated problem in the field of materials modelling [J. Maurer *et al.*, *Annu. Rev. Mater. Res.* 49, 1–30 (2019)].

Reviewer #2:

“This manuscript discusses results of the neutron scattering measurements of VO(acac)2. These results may be of interest to the part of the neutron scattering community specializing in phonons. This study is solid, and the results/interpretations are valid. However, I don't see any significant broader impact for a number of reasons.”

We thank the Reviewer for stating that our results/interpretations are valid. We respectfully disagree, when he says that the results are of interest mainly “to the part of the neutron scattering community specializing in phonons”. Indeed, we are not part of that community. We believe that Reviewer’s judgment could have been affected by some weaknesses in the presentation of our results and their consequences. We have now improved it and we reply to the detailed points below.

We would like also to underline that the interest of our results has been recognized by other Reviewers.

“1. The main finding of anti crossings between acoustic and optic phonons in a crystal prepared from these molecules can only happen in crystals where acoustic branches exist. It cannot occur in single molecules where there are no acoustic phonons. So the main result of this work is completely irrelevant to single-molecule cubits.”

Magnetic relaxation **in crystals** of molecular nanomagnets has been the pivotal point of most of the research in molecular magnetism in the last 25 years, as witnessed by hundreds of publications. However, phonons, the main actors causing relaxation, have never been directly and deeply experimentally studied in MNMs before. Hence, the present results are of great relevance for the whole molecular magnetism community.

Even if we limit to quantum information processing applications, the vast majority of the studies on molecule-based quantum computing are done on ensembles. Indeed, important advancements have been recently achieved in building magnetically diluted single crystals (see e.g. F. Moro, et al., Chem. Commun., 50, 91 (2014)). For instance, in a quantum simulation experiments, working with an ensemble is not only easier, but has the advantage of immediately yielding expectation values of the observables. All these applications will be in crystals with phonons similar to those here studied.

At last, present results are important even in the perspective of scalable devices where single molecules will be grafted on surfaces. Indeed, our results demonstrate the validity of DFT in predicting vibrational modes in these kind of systems. Hence, grounded on these results we could use DFT also to make prediction in single molecules.

We have now modified the paper to make the presentation and discussion clearer. In addition, we have added a paragraph on the “take-home message” for the synthesis of new molecular nanomagnets.

“2. The anticrossings reported here are generic and occur almost all the time, so the novelty of this work is not that high.”

As discussed above, this work represents the first experimental study of phonon dispersions in a molecular nanomagnet and ACs are only one of the findings. Hence, the novelty goes far beyond ACs.

However, the ACs between acoustic and optical phonons are probably common in molecular magnetism (where optical phonons can be expected to occur at low energies), but are not common of all materials. Indeed, often optical phonons are too high in energy for this kind of AC to occur. For instance, there has been an intense research for engineering thermo-electric materials to obtain this kind of ACs (see e.g., C. H. Lee, *et al.*, J. Phys. Soc. Jpn. 75, 123602 (2006), M. Christenses, *et al.*, Nat. Mater. 7, 811 (2008), E. S. Toberer, *et al.*, J. Mater. Chem. 21, 15843 (2011)).

Moreover, we performed new calculations showing the key effects of these ACs on spin-phonon couplings, to better discuss their link with relaxation dynamics.

We have significantly expanded the Discussion Section to report the new results.

“3. It would be more interesting to explore higher energy phonons, which are closer to what would be observed in actual qubits, but this has not been done due to limitations of the LET spectrometer.”

High-energy phonons are interesting, but are not typically involved in magnetic relaxation at low temperatures. Indeed, they don't contribute to resonant processes (due to their large energies). Moreover, such phonons don't contribute to Raman processes, because these are proportional to phonon populations, which are negligible for high-energy phonons at low temperatures. Hence, the dominant relaxation channels will anyway involve low-energy modes.

“4. As a relatively minor point, the agreement with the DFT calculations is not as good as claimed, since the phonons near the X-point are about 20% higher in energy than calculated, and the experimental anti crossing gap is much higher than the theoretical one.”

The agreement is actually good, by mistake previous Figure 5 did not contain the final results. We have now replaced it with the correct one. We thank the Referee for evidencing this problem.

Reviewer #3:

“The manuscript details the application of a new technique to studying phonons in a molecular qubit system. Vibrations, of molecules or their local matrices, are an intense focus in the world of molecular magnetism. In particular, understanding how vibrations correlate to magnetic relaxation time is a notable step toward the holy grail of finding. The manuscript by Garlatti and coworkers takes an important step toward that goal by using a new technique – 4D inelastic neutron scattering. They apply this technique for the first time to a molecular system, revealing evidence of phonon anticrossings between acoustical and optical branches. The application of the new technique to identifying phonons and how they specifically perturb a metal complex is significant and warranting publication.”

We are grateful to the Reviewer for recognizing the importance of understanding phonons and their role in magnetic relaxation of molecular nanomagnets, as well as the step towards this goal represented by our work. We also thank them for the comments and suggestions that we address here below.

“However, there are two aspects of the work that I think either require heavy revision or publication elsewhere. First, the manuscript is written in a way that renders it very difficult to access by those who could potentially use the work to design new molecular qubits by vibration tuning. I'd recommend some revision to help the synthetic folks out a little bit, especially if this is going into such a high-impact journal.”

We thank very much the Reviewer for the suggestion. Following their advice, we significantly edited the manuscript, with the aim to improve its accessibility to a broader audience. On the one hand, we worked keeping in mind the broad audience of Nat. Commun., but, on the other hand, we tried to maintain a balanced level of description for both physicists and chemists readers. We believe that our heterogeneous

backgrounds have finally helped us achieve this balance. Indeed, this work involved the collaboration of synthetic and computational chemists together with theoretical and experimental physicists.

First of all, we revised the “Results” section on 4D-INS: we moved the formulas and all the details about the scattering function in the “Methods” and we expanded the definition of transverse and longitudinal phonons. We also moved some details about DFT calculations from the “Introduction” to the dedicated section of the “Results”. We also improved the general readability of the paper by removing redundancies in the presentation of the results: We shortened the summary of the results at the end of the “Introduction” and we removed it from the beginning of the “Discussion” section.

Finally, in the “Discussion” section we added new results on the effects of low-lying optical modes and ACs on the relaxation dynamics (see following point) and we tried to summarize them into a strategy for the synthesis of future molecular systems.

“Second, the tie to magnetism (the most important implication) is stated in a manner that is too vague to make this an obvious high-impact publication. How exactly will the avoided crossings impact magnetism? Is it through controlling phonon lifetimes? How would the lifetime of a phonon or molecular vibration be expected to impact the spin relaxation times? There are a lot of manuscripts coming out attempting to tie vibrations to magnetism, and the foregoing questions are probably much too big to answer in one manuscript (and the given data do not answer all of these questions), but some additional development is necessary in my opinion. Without better explanation and context from the rest of literature (including a recent paper by Moseley et al. Eur. J. Inorg. Chem. 2019, 8, 1055), the impact of the manuscript seems inflated with no clearly articulated link between what was discovered/measured and an impact on magnetism.”

We are grateful to the Reviewer for highlighting this important weakness of our work. We followed his/her suggestion and we performed additional calculations in order to understand the effects of Anti Crossings (ACs) on magnetic relaxation in [VO(acac)₂].

Starting from our experimentally-tested DFT results and following the methods developed by A. Lunghi and S. Sanvito, arXiv:1903.01424 (2019), we have now calculated the spin-phonon coupling coefficients of low-energies phonon modes and studied their squared norm V^2_{SPh} (see new equation 1 in the main text) as a function of the phonon wavevector \mathbf{q} . We have focused on the phonon-induced modulation of the \mathbf{g} tensor, which is the main relaxation mechanism in this system at low temperatures (see arXiv:1903.01424 (2019)).

The new results, reported in the new Fig.7 and Fig.S2, show that at the ACs the significantly stronger spin-phonon coupling of the flat optical modes is partially "transferred" to the dispersive acoustic modes, which respectively display a minimum or a maximum of V^2_{SPh} . This mixing effect enhances the spin-phonon coupling of acoustic modes not only at the AC, but also at lower energies and \mathbf{q} , where they're still strongly dispersive. Thus, we expect that these enhanced acoustic modes could trigger effective relaxation processes by modulating the \mathbf{g} tensor with much larger probability and in a wider energy range. All these new results and observations have been added to the “Discussion” section of the paper.

As the Reviewer underlines, the role of the coupling between electronic spins and lattice vibrations and how it drives relaxation in molecular nanomagnets is a hot-topic of the current research in Molecular Magnetism. Thus, we contextualized more our research in the “Discussion” section by adding a new paragraph with references about previous works on the topic (Refs. 5, 18, 23, 32 and 56). Indeed, the main

aim of our work was to provide the first direct measurements of phonon dispersions and polarization vectors in a molecular nanomagnet. These are in fact the key ingredients for a quantitative evaluation of spin-phonon coupling coefficients and relaxation dynamics. We underlined that recent *ab initio* approaches applied to calculate spin-phonon couplings can't capture the full picture of spin relaxation since they only include normal modes in the gas-phase approximation or the full spectrum only at the Γ -point. In our work, we have also added quantitative results on spin-phonon coupling coefficients based on a very recent and rigorous method (A. Lunghi and S. Sanvito, arXiv:1903.01424 (2019)), which take into account the full description of lattice vibrations.

As far as phonon lifetimes are concerned, ACs between acoustical and optical phonons are known to enhance phonon scattering mechanisms and hence to reduce their lifetimes (see e.g., C. H. Lee, *et al.*, J. Phys. Soc. Jpn. 75, 123602 (2006), M. Christensen, *et al.*, Nat. Mater. 7, 811 (2008), E. S. Toberer, *et al.*, J. Mater. Chem. 21, 15843 (2011)). This in turn could further enhance magnetic relaxation (see e.g., A. Lunghi, *et al.*, Nat. Commun. 8, 14620 (2017)).

We also thank the Reviewer for pointing out the interesting works by Moseley *et al.*, Eur. J. Inorg. Chem. 8, 1119 (2019), which has been added to our references. In the there-cited Nat. Commun. 9, 2572 (2018)(Ref.12 of the paper) the authors experimentally observed the presence of spin-phonon couplings in a single-molecule magnet, since they induce avoided-crossings between electronic crystal field transitions and phononic excitations in field-dependent Raman spectra. It is important to stress that these ACs are different from the ones we observed here. Our ACs only involve phonon branches and exist regardless of the presence of spin-phonon couplings, even if they have a strong influence on them.

We have also cited the very recent Eur. J. Inorg. Chem. 8, 1055 (2019) as one interesting example where INS (even if non-4D) has been applied to study phonons in molecular nanomagnets. In addition, the authors also confirm and discuss the importance of taking into account van der Waals interactions to correctly reproduce low-energy vibrational spectra.

Reviewer #4:

“The paper provide convincing results on the interest of 4D-INS techniques coupled with DFT calculations to address the question of the impact of phonon on the relaxation processes in MNMs, and particularly the role of the AC between optical and acoustical branches. The experimental and theoretical parts are well presented and clear even for non specialist, the text is very clear. Then, it is highly relevant to publish it in Nat. Comm.”

We are grateful to the Reviewer for their very positive comments about the results of our manuscript and the clarity of the presentation. We also thank him/her for the suggestions, that we individually address below.

“About the impact of vdW interactions inducing “admixing between acoustic and optical branches”, I am wondering if DFT calculations using the same functional but without D3 corrections could be performed. Indeed, vdW interactions should be killed in these calculations and the comparison with the results presented in the paper could give more weight to the point above.”

The existence of low-energy optical modes (and of the corresponding ACs) is mostly due to the presence of soft coordination bonds. Moreover, the requirement of vdW-corrected functionals for the modelling of molecular crystals is by now confirmed and settled by several results in literature [J. Maurer *et al.*, *Annu. Rev. Mater. Res.* 49, 1–30 (2019), F. Brown-Altvater *et al.*, *Phys. Rev. B* 93, 195206 (2016), J. Hoja *et al.*, *WIREs Comput Mol Sci* 7, 1294 (2016), F. Tran *et al.*, *Phys. Rev. Mat.* 3, 063602 (2019), C. Motta and S. Sanvito, *J. Chem. Theory Comput.* 10, 4624 (2014)]. As stated in F. Tran *et al.*, *Phys. Rev. Mat.* 3, 063602 (2019), several vdW functionals have been proposed during the last few years due to the huge diversity of molecular and crystal structures and covalent and non-covalent interactions among molecular crystals. Thus, it is not always clear which vdW correction should be used together with the best software available for the system at hands. In this work, we measured, for the first time, phonon dispersions and polarization vectors in a molecular nanomagnet. These results allowed us to verify the reliability of phonon modes calculations (energies, dispersions and eigenvectors) on a molecular nanomagnet and in particular to test the cp2k software, which allows the 3x3x3 supercell with 1620 atoms calculation to converge in reasonable times, with the best available vdW-corrected functional PBE+D3.

Anyway, from an academic point of view, it might be interesting to perform also calculations without D3. However, DFT calculations are very heavy in this system and redoing them could delay too much the publication of our results. Indeed, in the time frame of the manuscript resubmission we have preferred to invest computational resources in the calculation of the q-dependence of the spin-phonon couplings, in order to highlight the effect of ACs on spin dynamics. The comparison of DFT calculations with and without the D3 correction could be the subject of a separate publication. We do expect a very similar scenario (with low-lying optical modes displaying ACs with the acoustic ones) due to the presence of soft coordination bonds, even without taking into account vdW corrections.

We added a comment and the aforementioned references about vdW corrections in the “Methods” section about DFT calculations.

“I suggest some simplification of the text:

**** “state-of-the-art DFT” appears 5 times in the text. I am convinced that the calculations have correctly been done and that the results are good enough to support the conclusions of the authors. But such repetition does not provide more proof and sounds like if authors were “self-satisfied” (I hope the meaning in English is clear, I am not sure of the translation, my aim is not to offend the authors).”***

****some sentences/ideas appears twice without, the text should then be lighten:***

- page 3 “the dot product”...“allows to select between longitudinal and transverse phonon modes” / “An important feature of INS technique is the possibility to experimentally distinguish between longitudinal and transverse modes by properly selecting the Q and phonon ...”

- the coupling between optical and acoustic branches and its impact is clearly presented, but I feel that authors repeat a bit too much this feature, as if they were not sure that it is clear to the reader.”

We thank the Reviewer for highlighting this redundancies: we limited the definition “state-of-the-art DFT” to the “Abstract” and “Introduction” of the manuscript and we reduced the discussion about the longitudinal/transverse phonon modes to the paragraph “An important feature of the INS technique...”. We left discussions about anti-crossings (ACs) between optical and acoustic branches only in some “strategic” parts of the text, trying to give them different points of view. We briefly anticipate the effects of ACs in the “Introduction”; we describe ACs in the experimental part of the “Results” section in order to motivate the

in-depth analysis of the data reported in Fig.4; we study the composition of normal modes at the ACs in the “DFT Simulations of 4D-INS data” section and in Fig.6; finally, we added a discussion about the effects of ACs on the spin-phonon coupling coefficients in the “Discussion” section and in Fig.7.

List of Changes:

Title

We added a reference to DFT calculations in the title and changed it from:

“Unveiling phonons in a molecular quantum bit with four-dimensional inelastic neutron”

to

“Unveiling phonons in a molecular quantum bit with four-dimensional inelastic neutron scattering and density functional theory”.

Abstract

- From line 7: We modified the second part of the Abstract in order to add a reference to our new results. We changed it from:

“Moreover, we evidence phonon anti-crossings involving acoustical and optical branches, which could be important for magnetic relaxation. Using energies and polarisation vectors calculated with state-of-the-art Density Functional Theory, we reproduce the energies and intensity patterns of the observed modes and reach a sound understanding of phonons in [VO(acac)₂]”

to

“Using energies and polarisation vectors calculated with cutting-edge Density Functional Theory, we reproduce the energies and intensity patterns of the observed modes and reach a sound understanding of phonons in [VO(acac)₂]. Moreover, we evidence phonon anti-crossings involving acoustical and optical branches, yielding significant transfers of the spin-phonon coupling strength between the different modes”.

Introduction

- Pag.1, Column 2, last line: we moved here a comment about the use of INS to measure phonons which was previously at the beginning of the section “Results – Unveiling phonons with 4D-INS”:
“Indeed, INS is a very powerful technique to study phonons, because it enables one to directly access phonon eigenvalues and eigenvectors. Moreover, the recent advent of spectrometers combining the time-of-flight technique with position-sensitive detectors makes measuring the 4-dimensional scattering function $S(\mathbf{Q}, \omega)$ in large portions of the reciprocal space possible.”
- Pag.1, Column 1, line 9: changed from “...dispersions in bulk systems...” to “...dispersions in inorganic systems with extended structures...”.

We summarised the last part of the Introduction on pag.2 Column 1, by changing and cutting several parts:

- Pag.2, Column 1, line 11: we changed from:
“This work focuses on a prototypical complex from the vanadyl (VO) family, VO-acetylacetonate, [VO(acac)₂]. VO-based molecular systems are in fact currently emerging as archetypes of a new generation of molecular qubits with long coherence times up to high temperatures. Beside this

scientific interest, the small size of [VO(acac)₂] unit cell and its relatively low number of atoms make it a promising system to reach a deep understanding of phonons in a MNM."

to

"This work focuses on VO- acetylacetonate ([VO(acac)₂]), a prototypical complex from the vanadyl (VO) family, archetype of a new generation of molecular qubits with long coherence times up to high temperatures."

- Pag.2, Column 1, line 15: we changed from:

"The results of this challenging experiment are compared to simulations based on state-of-the-art Density Functional Theory (DFT) calculations. It is worth to note that calculations of phonons for a generic Brillouin zone point by DFT is a daunting task, as the number of atoms that needs to be included is usually very large and approaches the computational limits of this method. Moreover, the accurate description of molecular crystals is particularly challenging due to the known shortcomings of DFT in describing van der Waals (vdW) interactions. The latter are expected to be particularly weak in [VO(acac)₂] which is highly volatile as most β-diketonate neutral complexes. In spite of these difficulties, our DFT results enable us to reproduce remarkably well 4D-INS data."

to

"The results of this challenging 4D-INS experiment are compared to simulations based on Density Functional Theory (DFT) calculations, which enable us to reproduce remarkably well 4D-INS data."

and moved the comments about DFT and vdW interactions in the section "Results - DFT Simulations of 4D-INS data".

- Pag.2, Column 1, line 31: we added a new comment about our latest results on spin-phonon couplings:

"Moreover, we have calculated ab initio the important effect of ACs on the spin-phonon coupling coefficients, key ingredients in determining the [VO(acac)₂] relaxation dynamics."

Results – The [VO(acac)₂] molecular qubit

- Pag.2, Column 2, last line of the section: we added the deuteration percentage of our samples: *"..with a high percentage of deuteration (ca. 98%)."*

Results – Unveiling phonons with 4D-INS

- We moved the first paragraph about INS in the "Introduction".
- We moved formulas 1 and 2 (now 3 and 4 respectively) and their description in the section "Methods - Data analysis and simulations".
- Pag.2, Column 2: the section now directly introduces the experimental results. We slightly changed the text from:

"We have exploited the cold-neutron LET spectrometer at ISIS, since..."

to

"We exploited the cold-neutron LET spectrometer at ISIS to measure the 4-dimensional scattering function $S(\mathbf{Q}, \omega)$ of [VO(acac)₂] (more details about $S(\mathbf{Q}, \omega)$ in the Methods sections), since..."

- Pag.3, Column 2, line 1: we added a brief explanation of phonon polarisations: *"...to experimentally distinguish between phonon modes with longitudinal or transverse polarisation, characterised respectively by lattice vibrations parallel or perpendicular to the direction of propagation. This is done by..."*

- End of the section: we removed the paragraph “The data reported in this section are examples of the intensity maps that can be extracted from the 4D dataset of our experiment...” to streamline the reading and the keep the connection with the next section.

Results - DFT Simulations of 4D-INS data

- Pag.5, Column 1, last line: the following comments about DFT and vdW interactions was moved here from the Introduction: *“It is worth noting that calculations of phonons for a generic Brillouin zone point by DFT is a daunting task, as the number of atoms that needs to be included is usually very large and approaches the computational limits of this method. Moreover, the accurate description of molecular crystals is particularly challenging due to the known shortcomings of DFT in describing van der Waals (vdW) interactions [49-51]. The latter are expected to be particularly weak in [VO(acac)2], which is highly volatile as are most β -diketonate neutral complexes.”*
- Pag. 6, Column 1, line 9: we refined the discussion about the energy rescaling, changing from: *“A small rescaling of 13% has been applied to lower the calculated phonon energies $\omega_j(\mathbf{q})$, due to the limitations of state-of-the-art DFT methods in describing vdW interactions in molecular systems, whose effects are especially visible at low energies. Nonetheless, the simulations...”* to *“small rescaling of 13% has been uniformly applied to all phonon modes, in order to lower the calculated phonon energies $\omega_j(\mathbf{q})$ and better reproduce INS data. This discrepancy is mainly due to the limitations of DFT methods in describing vdW interactions in molecular systems [49-51], especially at low energies. Nonetheless, the level of agreement with the INS data is consistent with other results in literature [52-54] and the simulations...”*.

Discussion

We rewritten this section by adding new results about spin-phonon coupling calculations and consequences on the relaxation dynamics. The new parts are:

- **Figure 7.**
- **From the beginning of the section on Pag. 6, Column 1 to Pag.7, Column 1, line 44.**
- Pag. 7, Column 2, line 9: we also added a summary of the consequences of our findings for the design of new systems: *“Our results point out that long coherence times or slow magnetic relaxation are undermined by the presence of acoustic-optical phonons ACs. Thus, future synthetic strategies should, for instance, reduce the presence of soft coordination bonds with the aim to shift optical branches to higher energies and thus remove some of the ACs. Few ACs could be also eliminated by increasing the crystal symmetry. However, this would only affect crossings along symmetry directions, while ACs would still be present along generic q directions.”*

Methods - DFT calculations

- Pag.8, Column 1, line 48: we added a comment about vdW corrections: *“Indeed, the requirement of vdW-corrected functionals for the modelling of molecular crystals is by now confirmed and settled by several results in literature [49-51, 53].”*

- Pag.8, Column 2, line 3: we added the details about the calculations of the spin-phonon coupling coefficients.

Methods - Data analysis and simulations

- Pag. 8, Column 2, beginning of the section: the formulas and the description of the one-phonon coherent scattering function was moved here from the section "Results - DFT Simulations of 4D-INS data".
- Pag. 9, Column 1, line 10: we added details about the DFT energy rescaling:
"To reproduce INS data a small rescaling of about 13% was uniformly applied to all the calculated phonon energies along all the symmetry directions."

References

We added references:

- 12: M. Atzori and R. Sessoli, J. Am. Chem. Soc. 141, 11339 (2019).
- 49: R. Maurer, C. Freysoldt, A. Reilly, J. G. Brandenburg, O. Hofmann, T. Bjorkman, S. Lebègue, and A. Tkatchenko, Annu. Rev. Mater. Res. 49, 1 (2019).
- 50: J. Hoja, A. Reilly, and A. Tkatchenko, WIREs Comput.Mol.Sci. 7, 1294 (2016).
- 51: F. Tran, L. Kalantari, B. Traore, X. Rocquefelte, and P. Blaha, Phys. Rev. Mat. 3, 063602 (2019).
- 54: S. E. Stavretis, Y. Cheng, L. L. Daemen, C. M. Brown, D. H. Moseley, E. Bill, M. Atanasov, A. J. Ramirez-Cuesta, F. Neese, and Z.-L. Xue, Eur. J. Inorg. Chem. 8, 1119 (2019).
- 56: L. Escalera-Moreno, N. Suaud, A. Gaita-Ariño, and E. Coronado, J. Phys. Chem. Lett. 8, 1695 (2017).

Reviewers' comments:

Reviewer #1 (Remarks to the Author):

In my opinion the authors have addressed my own initial reservations and those of the other referees with respect to the previously submitted manuscript. The revised manuscript is a comprehensive and thorough investigation of the phonon (low-energy molecular excitations) behaviour of an archetypal molecular nano magnet that remains computationally tractable. It is somewhat reassuring that the findings of the DFT computations make implicit sense - as such this manuscript adds some rigour to previous 'hand-waving' arguments. The significance of magnetisation relaxation pathways is obvious if the desire is to preserve the magnetisation (much of their interest in such materials as quantum or data-storage devices) - phonons are intrinsic to this.

I have no hesitation in recommending that the revised manuscript be accepted for publication in Nature Communications - it is well written and is accessible to a broad readership and will be of potential interest to a wide range of researchers.

I commend the authors for a job well-done.

Reviewer #2 (Remarks to the Author):

I would like to thank the authors for revising the manuscript and making it more clear. In particular, discussion and conclusions are well written and give a good reflection of the work as I understand it. The presentation of the material in the results section is unfortunately too ambiguous (see below) but at the very least, this work has shown that DFT can provide an adequate description of lattice dynamics in molecular nanomagnet crystals. As a consequence, the same calculation will give a similarly good description of a single molecule, but may not work very well for a clusters of molecular nanomagnets too small for long range order approximation to be made. Still, given the great interest in the field, perhaps this article may appeal to a general audience. The manuscript still has a number of shortcomings that would preclude its publication in the present form.

I recommend adding the discussion of single-molecule magnets vs. crystals in the reply to my original report to the introduction.

There is a claim that experimental phonon dispersions are in "perfect agreement" with Fig. 5. Also the paper claims that a comparison of the color plot simulations based on the DFT calculation and the color plots of the data show a very good agreement for the normal mode eigenvectors. Unfortunately it is not very easy to judge how good the agreement for either because of the way the data and calculation results are presented. In order to compare the calculated and experimental dispersions it is common to show them in the same plot with points representing experimental phonon energies and lines representing DFT results. I would like to see such a plot. Also, the color plots can look similar even when the intensity of the features differs by as much as a factor of 2. Knowing how the intensities compare is needed to see how well experimental and theoretical eigenvectors match. The best way to show that is to plot the raw data together with the prediction of the DFT. From looking at the color plots, I can say that acoustic phonon intensity below the AC region is well reproduced, but the intensity at higher energy that couple most strongly to the magnetic degrees of freedom is not. In particular, I would like to see the following constant Q cuts where data are superimposed with calculations: (0,2.25,-5), (0,2,-4.5), (-1,0,-2.7),(-1.5,0,-3),(-1,0,-2.8).

The authors claim that they calculated the effect of AC on the spin-phonon coupling coefficients. However it is not clear from the paper what the effect is. They show the strength of spin-phonon coupling in each of the phonon branch in fig. 7 only in crystals with long-range order, well-defined acoustic phonons and ACs. To separate out the effect of ACs, I would like to see the same

calculation done on a single molecule where no acoustic branches are present and there are no ACs. Comparison of the two calculations will isolate the effect of ACs. In particular, for the broad audience interested in quantum computing, it would be most interesting to see the difference in the quantum spin coherence calculated with and without ACs. If such calculations cannot be done, I recommend to change the wording. Better yet, to make the paper of greater interest to the general audience, I recommend adding these calculations to the paper if possible.

The authors seem to use the definition of the acoustic branch as the lowest energy branch, but this is not a standard definition to the best of my knowledge. Acoustic branch is the branch that most closely approximates sound waves. When there is an AC between the acoustic and an optic branch, the optic branch is the lower one, and acoustic is the higher one on the high q side of the AC. So the calculation in Fig. 7 actually shows that the spin-phonon coupling is always connected to the optic character. The acoustic branch acquires it only near the AC where it mixes with the optic branch, and then loses it when the two branches split apart at higher q . So no "transfer" of spin-phonon coupling occurs as stated in the paper except right at. I recommend to change the discussion/intro to reflect this.

Sentence on top of page 6 starting: "Despite these works provided..." is not well worded and I don't understand what it really means.

Reviewer #3 (Remarks to the Author):

I want to thank the authors of the paper for their time editing the paper. The context, motivation, and importance are all better explained and described. Furthermore, the results are clearly delineated, and I suspect the paper will inspire many others to try 4D-INS (it certainly will for my lab). I think the paper is nearly ready for acceptance, but I think it would be worth a few readovers to check for typos or awkward phrasings. For example, on page 2 at end of introduction it reads "deer" when should be "der".

And one more point: the vanadyl bond is commonly misrepresented as a double bond, when it is a triple bond (Acc. Chem. Res. 2018, 51, 1850–1857; Inorg. Chem. 1962, 1, 111–122). So a few replacements of "double" with "triple" would correct this point in the description of the complex of interest.

Reviewer #4 (Remarks to the Author):

All the comments I made have been addressed in the new version. I propose that it should be published without anymore change.

We would like to **thank all the Reviewers**, for appreciating our efforts in improving the paper. In particular, we would like to thank Reviewers 1, 3 and 4 for their very positive judgments like “I have no hesitation in recommending that the revised manuscript be accepted for publication in Nature Communications - it is well written and is accessible to a broad readership and will be of potential interest to a wide range of researchers”, “I suspect the paper will inspire many others to try 4D-INS (it certainly will for my lab)” or “I propose that it should be published without anymore change”. Reviewer 2 states “discussion and conclusions are well written and give a good reflection of the work”, but believes that “The manuscript still has a number of shortcomings that would preclude its publication in the present form”. Following Reviewer 2’ s advices, we have now **performed additional data analysis and new calculations that confirms our results** and further strengthen our conclusions.

Reviewer #1:

“In my opinion the authors have addressed my own initial reservations and those of the other referees with respect to the previously submitted manuscript. The revised manuscript is a comprehensive and thorough investigation of the phonon (low-energy molecular excitations) behaviour of an archetypal molecular nanomagnet that remains computationally tractable. It is somewhat reassuring that the findings of the DFT computations make implicit sense - as such this manuscript adds some rigour to previous 'hand-waving' arguments. The significance of magnetisation relaxation pathways is obvious if the desire is to preserve the magnetisation (much of there interest in such materials as quit or data-storage devices) - phonons are intrinsic to this.

I have no hesitation is recommending that the revised manuscript be accepted for publication in Nature Communications - it is well written and is accessible to a broad readership and will be of potential interest to a wide range of researchers.

I commend the authors for a job well-done.”

We are grateful to the Reviewer for his/her very positive comments on the importance of the results and the clarity of the presentation of our revised manuscript. We are glad to have addressed all their previous remarks and we thank for recommending the manuscript for publication in Nature Communications.

Reviewer #2:

“I would like to thank the authors for revising the manuscript and making it more clear. In particular, discussion and conclusions are well written and give a good reflection of the work as I understand it. The presentation of the material in the results section is unfortunately too ambiguous (see below) but at the very least, this work has shown that DFT can provide an adequate description of lattice dynamics in molecular nanomagnet crystals. As a consequence, the same calculation will give a similarly good description of a single molecule, but may not work very well for a clusters of molecular nanomagnets too small for long range order approximation to be made. Still, given the great interest in the field, perhaps this article may appeal to a general audience.”

We thank the Reviewer for recognizing our efforts to improve the paper. We are also grateful for his/her positive comments about the discussion and conclusions of the revised manuscript and about the significance of our results for a general audience.

“The manuscript still has a number of shortcomings that would preclude its publication in the present form.”

We addressed all the new Reviewer’s remarks to improve the presentation of our results. As detailed below, we have improved the comparison between the experimental results and the DFT calculations by **superimposing experimental data and calculations in Fig. 5** and adding **many 1D cuts** (again with calculations superimposed to data) in the Supplementary information. These figures **further strengthen our conclusions**. Moreover, we have **performed new DFT calculations** of the vibrational modes and magneto-vibrational couplings of an isolated [VO(acac)₂] molecule and compared them with phonon modes and spin-phonon couplings of the crystal. We have also added a discussion in the “Introduction” section about studying ensembles vs single-molecules and clarified the role of acoustic branches in ACs in the “Discussion” section. Please find below point-by-point answers and consequent changes and additions in the text.

“I recommend adding the discussion of single-molecule magnets vs. crystals in the reply to my original report to the introduction.”

We thank the Reviewer for the suggestion to add this discussion. It is now included in the “Introduction” section of the paper.

In particular, in the Introduction we have added: “While single molecules could be the long-term goal for applications, the majority of experimental studies on the relaxation dynamics of molecular nanomagnets were performed in crystals or polycrystalline samples. In addition, the use of magnetically diluted single crystals (see e.g. F. Moro, et al., Chem. Commun., 50, 91 (2014), now Ref. 33) can bring advantages in quantum simulation protocols, where the ensemble measurements immediately yield expectation values of the observables. Hence, the combination of an experimental technique directly addressing phonon dispersions and eigenvectors of a MNM crystal with state-of-the-art ab initio calculations is the starting point to investigate the physics behind relaxation mechanisms and benchmark theoretical models.”

“There is a claim that experimental phonon dispersions are in “perfect agreement” with Fig. 5. Also the paper claims that a comparison of the color plot simulations based on the DFT calculation and the color plots of the data show a very good agreement for the normal mode eigenvectors. Unfortunately it is not very easy to judge how good the agreement for either because of the way the data and calculation results are presented. In order to compare the calculated and experimental dispersions it is common to show them in the same plot with points representing experimental phonon energies and lines representing DFT results. I would like to see such a plot. Also, the color plots can look similar even when the intensity of the features differs by as much as a factor of 2. Knowing how the intensities compare is needed to see how well experimental and theoretical eigenvectors match. The best way to show that is to plot the raw data together with the prediction of the DFT. From looking at the color plots, I can say that acoustic phonon intensity below the AC region is well reproduced, but the intensity at higher energy that couple most strongly to the magnetic degrees of freedom is not. In particular, I would like to see the following constant Q cuts where data are superimposed with calculations: (0,2.25,-5), (0,2,-4.5), (-1,0,-2.7),(-1.5,0,-3),(-1,0,-2.8).”

We thank the Reviewer for highlighting this important aspect in the presentation of our work, which we have now improved. We never stated that “experimental phonon dispersions are in perfect agreement with Fig. 5”, we used “perfect agreement” only about the qualitative finding of low-energy optical modes and the corresponding ACs. Anyway, we recognize from Reviewer’s comments that this was not clear and we now expanded the comparison to make it clearer.

As suggested by the Reviewer, **we have modified Fig.5 by adding the energies of experimental phononic excitations extracted by fitting 1-dimensional cuts of the data reported in Figs.2 and 3, superimposed to DFT calculations**. This comparison further demonstrates the good agreement between the calculated

phonon energies and the experimental results. **Many representative 1-dimensional cuts, including those explicitly requested by the Reviewer, are now reported in Supplementary Note 2.** We have also added some comments about the new results reported in Fig.5 in the “Results - DFT Simulations of 4D-INS data” section and we described how we obtained 1-dimensional cuts in the “Methods - Data analysis and simulations” section.

We would like to stress that our theoretical results are obtained with *ab initio* calculations, i.e. with no fitting parameters (apart from a slight overall final rescaling of energies). Thus, we cannot expect experimental data and calculations to be perfectly superimposed (discrepancies are of the order of those found in literature for molecular crystals). **The new Fig.5 makes the comparison direct and clearer.**

We have also obtained a good description of the phononic excitations intensities, as shown by colour-maps in Figs.2 and 3 and **the new 1-dimensional cuts of Fig.S2 requested by the Reviewer.** Some discrepancies are evident in low-intensity excitations, where it is more difficult to isolate signal from background. Hence, as stated also by the Reviewer, we do think “*DFT can provide an adequate description of lattice dynamics in molecular nanomagnet crystals*”.

At last, we would like to stress that there are no similar works on molecular nanomagnets yielding the **same level of description of phonon energies, dispersions and eigenvectors** to compare our results with.

“The authors claim that they calculated the effect of AC on the spin-phonon coupling coefficients. However it is not clear from the paper what the effect is. They show the strength of spin-phonon coupling in each of the phonon branch in fig. 7 only in crystals with long-range order, well-defined acoustic phonons and ACs. To separate out the effect of ACs, I would like to see the same calculation done on a single molecule where no acoustic branches are present and there are no ACs. Comparison of the two calculations will isolate the effect of ACs. In particular, for the broad audience interested in quantum computing, it would be most interesting to see the difference in the quantum spin coherence calculated with and without ACs. If such calculations cannot be done, I recommend to change the wording. Better yet, to make the paper of greater interest to the general audience, I recommend adding these calculations to the paper if possible.”

Following the Reviewer suggestion, we have performed **new DFT calculations of the vibrational modes and the magneto-vibrational couplings of a single [VO(acac)₂] molecule**, with the same CP2K software used for the single crystal calculations. Details about these new calculations have been added to the “Methods - DFT calculations” section. Moreover, the lowest vibrational energies and the corresponding magneto-vibrational couplings are reported in Table S1 (Supplementary Note 4). Clearly, vibrational modes of an isolated molecule do not show dispersion and can only be compared with single crystal ones **at the Γ point** (where no acoustic-optical phonon ACs can occur). The comparison shows a substantial shift to lower energies of the first molecular mode with respect to the first phonon frequency at Γ (from about 4.8 meV to 3 meV), corroborating the non-negligible effect of crystal packing already at the level of a Γ -point description of the phonons. There is a **good correspondence between the molecular displacements associated with the low-energy isolated-molecule modes and those of optical phonons at the Γ point**, leading to similar values of the calculated magneto-vibrational coupling coefficients with respect to spin-phonon ones.

As far as relaxation is concerned, it is important to remark that, while the crystal provides a continuous spectrum of phonons at low energies, the isolated molecule does not. This implies that, starting from isolated-molecule vibrational modes and considering direct processes, it is not possible to predict a finite spin relaxation time, unless a strict resonance condition is reached between the molecule energy gaps and the accessible vibrational modes. Thus, this condition prevents direct processes to occur for reasonable values of the applied magnetic fields, suppressing the main relaxation mechanism and yielding **longer coherence times for the isolated molecules than in single crystal.**

This comparison does not enable to single out the role of ACs, because they do not occur at the Γ point. However, the effect of the ACs is already clear from the calculations in Figure 7. Indeed, it shows that the spin-phonon coupling of acoustic modes strongly increases when the degree of admixing with the optical one rises, i.e., when ACs are approached. This point is also recognized by Reviewer 2 (see his next remark). Hence,

if ACs are removed by chemical engineering in order to increase the energy of optical modes, the relaxation time will increase (because relaxation is dominated by low-energy modes).

Results of new DFT calculations in gas-phase are reported in Supplementary Note 4 and compared with Γ point optical phonons. These results are quoted in the Discussion section. In addition, details about these new calculations have been added to the “Methods - DFT calculations” section.

“The authors seem to use the definition of the acoustic branch as the lowest energy branch, but this is not a standard definition to the best of my knowledge. Acoustic branch is the branch that most closely approximates sound waves. When there is an AC between the acoustic and an optic branch, the optic branch is the lower one, and acoustic is the higher one on the high q side of the AC. So the calculation in Fig. 7 actually shows that the spin-phonon coupling is always connected to the optic character. The acoustic branch acquires it only near the AC where it mixes with the optic branch, and then loses it when the two branches split apart at higher q . So no “transfer” of spin-phonon coupling occurs as stated in the paper except right at. I recommend to change the discussion/intro to reflect this.”

We do agree with the Reviewer: acoustic branches are not defined as the lowest in energy, they are those that tend to zero at the Γ point. We also agree with the Reviewers that after the AC the optical branch is at an energy lower than that of the acoustic one. Most importantly, the Reviewer is right, the “transfer” of spin-phonon coupling is connected with the mixing of optical and acoustical modes and **it is maximum at the AC**. However, Figure 7 actually shows that the “transfer” of spin-phonon coupling **is not negligible even well before the AC, at lower $|q|$ values and lower energies**, where the acoustic mode is still strongly dispersive. **Indeed, the colour of the involved acoustic mode start to change at energies significantly smaller than the AC one.**

It is well known that low-energy dispersive modes are crucial for magnetic relaxation. Hence, the small optical component present in the nominally acoustic mode at low energies can significantly influence magnetic relaxation, by modulating the g tensor with much larger probability and in a wider energy range than without the mixing of modes, which is evidenced at higher energies by the ACs.

Probably **this point was not sufficiently clear** and we have **now improved the description of Figure 7** in the “Discussion” section of the manuscript.

“Sentence on top of page 6 starting: “Despite these works provided...” is not well worded and I don’t understand what it really means.”

The sentence on page 6, starting with “Despite these works provided...”, concerns previous works (e.g., Refs.[5, 23, 32, 58] of the present version of the manuscript) where *ab initio* calculations have been used to predict the individual contribution of vibrational modes to the spin-phonon coupling in magnetic molecules. These works provided an advancement in the understanding of spin relaxation mechanisms in molecular nanomagnets, but still they obtained a limited agreement with experimental results. This is due to limitations of the adopted computational models. For instance, calculations in these papers were performed in the gas-phase approximation or only included phonons at the Γ point.

We have now modified the sentence to improve its clarity: “Recently *ab initio* calculations have been used to predict the contribution of individual vibrational modes to the spin-phonon coupling [23, 32, 58]. The agreement with experimental results was limited because only vibrations at the Γ -point were included [32] or due to the gas-phase approximation [5].”

Reviewer #3:

“I want to thank the authors of the paper for their time editing the paper. The context, motivation, and

importance are all better explained and described. Furthermore, the results are clearly delineated, and I suspect the paper will inspire many others to try 4D-INS (it certainly will for my lab). I think the paper is nearly ready for acceptance, but I think it would be worth a few readovers to check for typos or awkward phrasings. For example, on page 2 at end of introduction it reads “deer” when should be “der”.

We thank the Reviewer positive comments about the significance and the clarity of our revised manuscript. We double-checked the text with the aim to improve the phrasing and correct all the typos (e.g. “deer” instead of “der” at the very end of the Introduction section). Please find below (after the response to Reviewer #4) a complete list of all the changes.

“And one more point: the vanadyl bond is commonly misrepresented as a double bond, when it is a triple bond (Acc. Chem. Res. 2018, 51, 1850–1857; Inorg. Chem. 1962, 1, 111–122). So a few replacements of “double” with “triple” would correct this point in the description of the complex of interest.”

We do agree with the Reviewer that a multiple bond character stronger than 2 can be found in the V-O bond for the vanadyl ion in vanadyl bis-acetylacetonate. Nonetheless, the majority of structural data reported so far in the literature report structures where the bond is represented as double bond. It should be considered that this character is not arbitrarily assigned, but it is the result of the evaluation of the bond length during crystal structure resolution. The paper *Inorg. Chem. 1962, 1, 111–122* is actually focused on the determination of electronic orbital level diagram for the vanadyl ion in solution (as an aquo complex) and its capability to be preserved in solution upon protonation, a property that we do not argue. The *Acc. Chem. Res. 2018, 51, 1850–1857* review paper cite the previous article at the very beginning of the introduction, but do not provide further experimental evidence for a triple bond in vanadyl metallorganic complexes, being the article more focused on reviewing oxo-iron species with important roles in biochemistry.

We added a footnote on this point in the “Results – The [VO(acac)₂] molecular qubit” section of the paper to cite that a triple bond character of the V-O bond in vanadyl has been proposed.

Reviewer #4:

“All the comments I made have been addressed in the new version. I propose that it should be published without anymore change.”

We are pleased to have addressed all Reviewer’s comment in our revised manuscript and we thank him for recommending it for publication in Nature Communications.

List of Changes:

Introduction

- Pag.1, Column 2, line 23. we have added a **new discussion** about single molecules and crystals and the new reference 33: *“While single molecules could be the long-term goal for applications, the majority of experimental studies on the relaxation dynamics of molecular nanomagnets were performed in crystals or polycrystalline samples. In addition, the use of magnetically diluted single crystals [42] can bring advantages in quantum simulation protocols, where the ensemble measurements immediately yield expectation values of the observables. Hence, the combination of an experimental technique directly addressing phonon dispersions and eigenvectors of a MNM crystal with state-of-the-art ab initio calculations is the starting point to investigate the physics behind relaxation mechanisms and benchmark theoretical models.”*

- Pag.2, Column 1, line 24: we changed from “...from the vanadyl (VO) family...” to “embedding the vanadyl (VO) unit”.
- Pag.2, Column1, last paragraph of the section: we removed the long paragraph with the summary of the results presented in the “Discussion” section, in order to remove redundancies and shorten the section. We added a brief comment: “Moreover, *ab initio* calculations have evidenced that ACs between acoustic and optical phonons can strongly affect the spin dynamics.”

Results - The [VO(acac)₂] molecular qubit

- Pag.2, Column 1, line 4. we have added **footnote 43** discussing the character of the bond within the VO²⁺ ion: “It should be highlighted that a triple bond has been proposed to describe the chemical bond in VO²⁺ (see C.J. Ballhausen and H.B. Gray, *Inorg. Chem.* 1, 111-122 (1962) and A.H.B. Gray and J.R. Winkler, *Acc. Chem. Res.* 51, 1850-1857 (2018)) despite it is commonly described as a double bond. A partial triple bond character has been evidenced through theoretical calculation by analyzing the electronic structure of [VO(acac)₂] [44], which is the subject of the present study.”

Results - DFT Simulations of 4D-INS data

- Pag.5, Column 2, Line 12. We have added a **discussion on the new results reported in Fig.5**: “...,in agreement with the experimental results. Indeed, Fig.5 also show the energies of some representative phononic excitations extracted from the data reported in Figs.2 and 3 superimposed to DFT calculations. This comparison demonstrates the very good agreement between the calculated phonon energies and the experimental findings. Only a small rescaling of 13% has been uniformly applied to all phonon modes, in order to lower the calculated phonon energies $\omega_j(\mathbf{q})$ and better reproduce INS data. The observed discrepancies are consistent with other results in literature for molecular crystals [54-56] and are mainly due to the limitations of DFT methods in describing vdW interactions in these systems [51-53], especially at low energies.”
- Pag.6, Fig.5 and caption. **new data analysis** have been performed and results are shown in the **new Fig.5**. As reported in the new caption, Fig.5 shows: “**DFT calculations of phonon dispersions compared to experimental excitation energies**. DFT calculations of the low-energy phonon dispersion curves of [VO(acac)₂] up to 15 meV along the path X(0.5,0,0)- Γ (0,0,0)-Y(0,0.5,0)- Γ (0,0,0)-Z(0,0,0.5) (black scatters), where the coordinates are expressed in the reciprocal lattice basis units. Red scatters are the energies of experimental excitations extracted by fitting 1-dimensional cuts of the data reported in Figs.2 and 3 (some representative cuts are reported in Supplementary Note 2, error bars are of the order of scatters dimension). Not all the modes are visible in the explored configurations. Blue circles high-light ACs between acoustic and low-lying optical branches along Γ -X and Γ -Z. Red and green dashed lines indicate the q vectors at which we have investigated the atomic displacements before and at the ACs, respectively.”

Discussion

- Pag.7, Column 1, Line 9. We have improved the clarity of the discussion about the results reported in Fig.7: “However, at the ACs the stronger spin-phonon coupling of the almost flat optical modes is partially “transferred” to the dispersive acoustic modes, which display a maximum of V^2_{SPh} . This is evident for both the AC involving the transverse acoustic mode analysed in Figs.4 and 6 and for the AC involving the longitudinal one at higher energies. Figure 7 actually shows that the mixing of the modes is not negligible even well before the AC, at lower $|q|$ values and lower energies, where the acoustic mode is still strongly dispersive. Indeed, the enhancement of spin-phonon coupling for the

nominally acoustic mode starts at energies significantly smaller than the AC one. It is well known that low-energy dispersive modes are crucial for magnetic relaxation. Hence, the small optical component present in the nominally acoustic mode at low energies can significantly influence magnetic relaxation by modulating the g tensor with larger probability and in a wider energy range than without the mixing of modes.”

- Pag.7, Column 2, Line 31. We have clarified the sentences about previous *ab initio* results and added a reference to the new isolated-molecule results reported in Supplementary Note 4: “*Recently ab initio calculations have been used to predict the contribution of individual vibrational modes to the spin-phonon coupling [23, 32, 58]. The agreement with experimental results was limited because only vibrations at the Γ -point were included [32] or due to the gas-phase approximation [5]. For comparison, gas-phase calculations of magneto-vibrational couplings for the present molecule (see Methods) have been performed and are reported in Supplementary Note 4.*”
- Pag.7, Column 1, line 42. We simplified the sentence, changing it from: “*...the strong mixing of the two modes involved in the ACs not only influences acoustic phonons group velocity but also phonon lifetimes, which both decrease at the ACs area.*” to “*the strong mixing of the two modes involved in the ACs decreases acoustic phonons group velocity and also phonon life-times.*”
- Fig.7 on Pag.7. We added the missing label on the x-axis.

Methods - DFT calculations

- Pag. 8, Column 2, Line 3. We have added details about DFT calculations on the single molecule: “*The optimization of the isolated molecule has been done with a plane-wave cutoff of 850 Ry and by turning off the periodic boundary conditions. An integration step of 0.01 Å was used for the calculation of the isolated molecules force constants.*”

Methods - Data analysis and simulations

- Pag. 9, Column 1, Line 12. We have added details about how we obtained and analysed 1-d cuts of the 4D-INS data: “*1-dimensional cuts as a function of energy were obtained by integrating 2D slices over the surviving direction in an interval centred around the desired Q value. They were then fitted with Gaussian functions in order to extract excitation energies in Fig.5. Representative examples are reported in Supplementary Note 2 with the corresponding simulated curves.*”

Supplementary Information

- Pag.1. We renamed Supplementary Note 1 Supplementary Figure 1, according to Nat. Commun. Guidelines.
- Pag.2. We have added a **new Supplementary Figure 2**, with representative examples of 1-dimensional cuts of the 4D-INS data as a function of energy, obtained by integrating 2D maps in Figs. 2 and 3 over the surviving direction in an interval centred around the desired Q value.
- Pag.3. We renamed Supplementary Note 3 Supplementary Figure 3, according to Nat. Commun. Guidelines.
- Pag.3. We have added a **new Supplementary Note 4, with the new Table S1**. Here we report the **new DFT results** on the lowest vibrational energies and the corresponding magneto-vibrational couplings of a single [VO(acac)₂] molecule, compared with energies and spin-phonon couplings obtained for the single crystal.

Reviewers' comments:

Reviewer #2 (Remarks to the Author):

The new version of the manuscript greatly improved presentation of the data and made the comparison with theory more clear. These results are interesting and worth publishing. Examining these figures shows that DFT calculations reproduce experimental dispersions only qualitatively (Fig. 5). More importantly, there are important disagreements between calculated and experimental dispersion in the anticrossing regions around the blue circles. For example, the dispersions from the Gamma point to the Z point are completely off and the experimental anticrossing gap is much larger than predicted.

New figures shown in Supplementary information (Fig. S2) show that the read curve, deviates significantly from the data more than half of the time. The peak positions corresponding to the red curve represent phonon energies, and the intensities are determined by eigenvectors. In the case of optic phonons, the intensities are off in the majority of cases and peak positions are off by more than 50% of the time. These plots represent a more accurate representation of how the data compare with theory than color plots.

Since the DFT does not reproduce eigenvectors accurately, it also will not reproduce spin phonon coupling, which is determined by these eigenvectors, so this is an important point.

These are serious problems with the way the paper is written, and therefore I conclude that the paper cannot be published as it is written.

However, I do believe that the raw data and raw calculation results are valuable and the scientific community will benefit from seeing them in a publication. So I am open to changing my mind on the recommendation not to publish this paper if:

1. The authors include figure S2 into the main body of the paper.
2. Move the color plots that obscure substantial disagreements between theory and experiment to supplementary information.
3. Change their conclusions and state clearly that the DFT calculations can reproduce ONLY some qualitative features in the phonon dispersions and need to be further improved to correctly reproduce spin-phonon interaction which requires accurate eigenvectors.

Response to the Reviewer

>The new version of the manuscript greatly improved presentation of the data and made the comparison with theory more clear. These results are interesting and worth publishing.

We would like to thank the Reviewer for his efforts in helping us to improve our paper and for recognizing the interest of our results.

>Examining these figures shows that DFT calculations reproduce experimental dispersions only qualitatively (Fig. 5). More importantly, there are important disagreements between calculated and experimental dispersion in the anticrossing regions around the blue circles. For example, the dispersions from the Γ point to the Z point are completely off and the experimental anticrossing gap is much larger than predicted. New figures shown in Supplementary information (Fig. S2) show that the read curve, deviates significantly from the data more than half of the time. The peak positions corresponding to the red curve represent phonon energies, and the intensities are determined by eigenvectors. In the case of optic phonons, the intensities are off in the majority of cases and peak positions are off by more than 50% of the time. These plots represent a more accurate representation of how the data compare with theory than color plots.

We have used state-of-the-art DFT, but we agree with the Referee that while qualitative features of experimental results are reproduced, a quantitative reproduction of both eigenvalues and eigenvectors requires further improvement of DFT calculations. We have now changed the abstract, the text and the conclusions accordingly (see below).

>Since the DFT does not reproduce eigenvectors accurately, it also will not reproduce spin phonon coupling, which is determined by these eigenvectors, so this is an important point.

We agree that spin phonon couplings are not quantitatively reproduced. However, important qualitative aspects like the presence of anticrossings and the non-negligible spin-phonon coupling extending to the lowest part of the acoustic branch depend on general features and are independent on the accuracy of the predictions.

> These are serious problems with the way the paper is written, and therefore I conclude that the paper cannot be published as it is written. However, I do believe that the raw data and raw calculation results are valuable and the scientific community will benefit from seeing them in a publication. So I am open to changing my mind on the recommendation not to publish this paper if:

>1. The authors include figure S2 into the main body of the paper.

We agree that 1D cuts enable one to directly judge the degree of agreement/disagreement between ab-initio experimental results and ab-initio calculations. Hence, we have now moved Figure S2 to the main text (it is the new Figure 2).

>2. Move the color plots that obscure substantial disagreements between theory and experiment to supplementary information.

We agree that color plots can obscure disagreements between theory and experiment, but this is now solved by the new figure 2 in the main text. We believe 2D slices of the 4D datasets provide an immediate visualization of phonon dispersions and demonstrate the capabilities of the 4D-INS experimental technique (as recognized also by the other Referees). Hence, we would prefer to keep also them in the main paper.

> 3. Change their conclusions and state clearly that the DFT calculations can reproduce ONLY some qualitative >features in the phonon dispersions and need to be further improved to correctly reproduce spin-phonon >interaction which requires accurate eigenvectors.

We also agree that while qualitative features of experimental results (like the presence of low-energy optical modes) are reproduced, a full quantitative reproduction of both eigenvalues and eigenvectors will require further improvement of DFT to treat vdW interactions. Therefore, we have changed the conclusions, the abstract and several statements in the paper. For instance, now in the “DFT Simulations of 4D-INS data” section we state that that “*quantitative discrepancies are evident especially from Fig.2.*”, and in the Conclusions we state “*We showed that while qualitative features of experimental results are reproduced, a full quantitative reproduction of both eigenvalues and eigenvectors will require further improvement of DFT to treat vdW interactions.*” The full list of changes is reported below.

List of Changes:

Abstract

- Line 7. We changed **from:** “*...calculated with cutting edge Density Functional Theory, we reproduce the energies and intensity patterns of the observed modes and reach a sound understanding of phonons in [VO(acac)₂].*” **to** “*...calculated with state-of-the-art Density Functional Theory, we reproduce important qualitative features of [VO(acac)₂] phonon modes, such as the presence of low-lying optical branches.*”
- We removed the last sentence.

Introduction

- Pag.2, Column 1, line 23. we changed **from** “*The results of this challenging 4D-INS experiment are compared to simulations based on Density Functional Theory (DFT) calculations, which enable us to reproduce remarkably well the 4D-INS data. Moreover, ab initio calculations have evidenced that ACs between acoustic and optical phonons can strongly affect the spin dynamics.*” **to** “*The results of this challenging 4D-INS experiment are compared to Density Functional Theory (DFT) calculations. Simulations reproduce important qualitative features of the 4D-INS data, such as the presence of low-lying optical modes and ACs between acoustic and optical phonons, providing insights on their effect on spin-phonon couplings.*”

Results - Unveiling phonons with 4D-INS

- Pag.3, Column 2, line 5. We added the sentence “*...to slice the 4D datasets into 1D curves to visualize excitations intensities as a function of energy around desired Q values*”, to introduce the possibility to visualize 1D cuts of the data (see next point).
- Pag.3, Column 1, Fig.2. We **moved Fig.S2 from the SI to the main text**, which is now the new Fig.2.
- Pag.3, Column1, caption of Fig.2. We **improved the clarity of the caption**, changing it **from** “*1-dimensional cuts. Black scatters: Representative 1-dimensional cuts as a function of energy obtained by integrating 2D maps in Figs. 2 and 3 over the surviving direction in an interval centred around the*

desired Q value. For data obtained with $E_i = 7.3$ meV the intensity is normalized for the maximum in each panel, while for data obtained with $E_i = 13$ meV the intensity is normalized in order to saturate acoustic modes and give prominence to optical ones. Red lines: Simulated intensity curves as a function of energy calculated with equation (3) of the main text and with phonon energies and polarisation vectors obtained from DFT.” to “Phononic excitations intensities vs energy. Black scatters: Representative examples of $[\text{VO}(\text{acac})_2]$ phononic excitations as a function of energy, obtained from LET data with $E_i = 7.3$ meV (a) and $E_i = 13$ meV (b) at $T = 5$ K. The 1D cuts have been obtained by integrating the 4D datasets over the three components of the neutron scattering vector $Q = [\eta; \zeta; \xi]$ (expressed in terms of reciprocal lattice vector units) around the desired values. For data obtained with $E_i = 7.3$ meV the intensity is normalized for the maximum in each panel, while for data obtained with $E_i = 13$ meV the intensity is normalized in order to saturate acoustic modes and give prominence to optical ones. Red lines: Intensity curves as a function of energy calculated with equation (3) (see Methods) and with phonon energies and polarisation vectors obtained from DFT.”

- Pag.3, Column 2, line 31. We added a discussion about the new Fig.2 “Phononic excitations intensities of $[\text{VO}(\text{acac})_2]$ are reported as a function of energy in Fig.2 for some representative Q values along Γ -X, Γ -Y and Γ -Z symmetry directions, for both $E_i = 7.3$ meV (Fig.2-a) and 13 meV (Fig.2-b) incident energies. The former show one or two excitation peaks, due to longitudinal and transverse acoustic modes, while $E_i = 13$ meV data show phononic excitations up to 11 meV from both acoustic and optical modes. 1D cuts give a detailed insight onto excitations, whereas 2D slices of the 4D datasets provide an immediate visualization of phonon dispersions, also demonstrating the capabilities of the 4D-INS technique. For instance,...”.
- Pag.3, Column 2, line 45. We changed from “...provide a full picture of $[\text{VO}(\text{acac})_2]$ acoustic modes up to 6 meV.” to “...provide a picture of $[\text{VO}(\text{acac})_2]$ acoustic phonons dispersions up to 6 meV”.
- Pag.4, Column 1, line 2. We changed from “..., we have measured with $E_i = 13$ meV (Fig.3)” to “..., we also report phonon dispersions obtained with $E_i = 13$ meV in Fig.4.”.

Results - DFT Simulations of 4D-INS data

- Pag.5, Column 2, Line 12. We changed from “...extracted from the data reported in Figs.2 and 3...” to “extracted from 1D cuts as in Fig.2...”.
- Pag.5, Column 2, Line 13. We changed the discussion about the agreement of the calculated phonon energies with the data from “This comparison demonstrates the very good agreement between the calculated phonon energies and the experimental findings. Only a small rescaling of 13% has been uniformly applied to all phonon modes, in order to lower the calculated phonon energies $\omega_j(\mathbf{q})$ and better reproduce INS data.” to “This comparison shows that calculated phonon energies reproduce important qualitative features of the experimental findings, such as the presence of low-lying optical modes and ACs with acoustic modes. However, a rescaling of 13% has been uniformly applied to all phonon modes, in order to lower the calculated phonon energies $\omega_j(\mathbf{q})$ and better reproduce INS data.”
- Pag.5, Column 1, Line 17. We changed the discussion about the agreement of the calculated 4D-INS spectra from “The simulations of 4D-INS spectra are reported in Fig.2-e-h and Fig.3-e-h and very well reproduce all the observed intensity patterns. In particular, while phonon dispersions are symmetric with respect to the Brillouin zone centre, neutron intensity patterns are not, as experimentally observed. Thus, being able to obtain high-quality experimental data for a direct comparison with simulations, we can state that our DFT results yield a reliable and sound description of phonon dispersions in $[\text{VO}(\text{acac})_2]$. DFT calculations can be further exploited to provide information on the

atomic motions associated with specific phonons.” **to** “*The simulations of 4D-INS spectra are reported in Fig.2 (red lines) for intensity vs energy 1D plots and in Fig.3-e-h and Fig.4-e-h for 2D colour-maps. Our calculations show an overall qualitative agreement with the observed excitations and dispersions. In particular, it is evident from Figs.3 and 4 that while phonon dispersions are symmetric with respect to the Brillouin zone centre, neutron intensity patterns are not, as experimentally observed. However, quantitative discrepancies are evident especially from Fig.2. Given that DFT results are able to reproduce important qualitative features of experiments, we can further analyse them in order to extract the atomic motions associated with specific phonons.*”

Discussion

- Pag.7, Column 1, Line 4. We changed **from** “*The here-determined phonons energies and eigenvectors enables a first quantitative calculation of the relaxation dynamics in this prototype molecular qubit. Indeed, a first promising ab initio simulation of direct relaxation processes has just been performed [57]*” to “*Indeed, the here-calculated phonons energies and eigenvectors enables the evaluation of the relaxation dynamics in this prototype molecular qubit. A first ab initio simulation of direct relaxation processes has just been performed [57].*”
- Pag.8, Column 1, Line 22. We **changed the final conclusions from** “*At last, the present study represents an important test for vdW-corrected DFT in describing phonon dispersions in this important class of materials. Indeed, the commonly used benchmark in this field is represented by calculated molecular crystals' cohesive energy [59, 60], whose direct comparison with experimental estimation is known to represent a challenge, due to the presence of multiple effects contributing to the measured values, (e.g., finite-temperature and zero-point-energy effects [61]).*” **to** “*At last, the present study represents an important test for vdW-corrected DFT in describing phonon dispersions. We showed that while qualitative features of experimental results are reproduced, a full quantitative reproduction of both eigenvalues and eigenvectors will require further improvement of DFT to treat vdW interactions. The DFT community involved in this field commonly use molecular crystals cohesive energy [59, 60] as benchmark test, whose direct comparison with experimental estimation is known to represent a challenge, due to the presence of multiple effects contributing to the measured values, (e.g., finite-temperature and zero-point-energy effects [61]).*”

Methods - Data analysis and simulations

- Pag. 9, Column 2. We added details about how we obtained and analysed 1-d cuts of the 4D-INS data:
line 14: “*...and into 1D curves to visualize excitations intensities as a function of energy around desired Q values.*”
- line 28: “*...in a ± 0.1 interval centred around the desired Q value and by applying a 7 bins smoothing, with error bars representing the SE.*”
line 39: “*For the simulation of 1D cuts in Fig.2 we also added the contribution of the elastic signal.*”

Supplementary Information

- **Supplementary Figure 2 has been moved to the main text** and it's now the new Fig.2.

REVIEWERS' COMMENTS:

Reviewer #2 (Remarks to the Author):

The authors incorporated my recommendations from the last round. I recommend the paper for publication as is.